# Alpha-synuclein stepwise aggregation reveals features of an early onset mutation in Parkinson's disease

Guilherme A.P. de Oliveira [1,2]* & Jerson L. Silva [1]*

Amyloid formation is a process involving interconverting protein species and results in toxic oligomers and fibrils. Aggregated alpha-synuclein (αS) participates in neurodegenerative maladies, but a closer understanding of the early αS polymerization stages and polymorphism of heritable αS variants is sparse still. Here, we distinguished αS oligomer and protofibril interconversions in Thioflavin T polymerization reactions. The results support a hypothesis reconciling the nucleation-polymerization and nucleation-conversion-polymerization models to explain the dissimilar behaviors of wild-type and the A53T mutant. Cryo-electron microscopy with a direct detector shows the polymorphic nature of αS fibrils formed by heritable A30P, E46K, and A53T point mutations. By showing that A53T rapidly nucleates competent species, continuously elongates fibrils in the presence of increasing amounts of seeds, and overcomes wild-type surface requirements for growth, our findings place A53T with features that may explain the early onset of familial Parkinson's disease cases bearing this mutation.

---

[1] Institute of Medical Biochemistry Leopoldo de Meis, National Institute of Science and Technology for Structural Biology and Bioimaging, National Center of Nuclear Magnetic Resonance Jiri Jonas, Federal University of Rio de Janeiro, Rio de Janeiro RJ 21941-901, Brazil. [2] Department of Biochemistry and Molecular Genetics, University of Virginia, Charlottesville, VA 22904, USA. *email: gaugusto@bioqmed.ufrj.br; jerson@bioqmed.ufrj.br

Alpha-synuclein (αS) aggregation participates in neurodegenerative maladies including Parkinson's disease (PD), multiple system atrophy, and dementia with Lewy bodies, but there is no consensus on what mechanisms trigger aggregation, neuronal cell loss, and degeneration. αS aggregates form toxic oligomers and amyloid filaments named Lewy bodies[1,2]. The presence of a particular αS strain causing multiple system atrophy[3], the seeding behavior, and the amplification of specific αS strains[4] dominate the prion concept in neurodegeneration[5].

Although most cases of synucleinopathies are sporadic, PD histories revealed familial variants of αS (e.g., A30P, E46K, and A53T) leading to early onsets of PD[6,7] and unique effects on their growth kinetics[8]. The A53T mutation was first reported in an Italian kindred and in three unrelated Greek families with autosomal dominant inheritance for PD[6]. The A30P was identified in three individuals of German origin[7]. All individuals presented an onset of illness ranging from the mid-30s to the mid-50s. The disordered nature of αS provides a framework to explore environmental conditions on filament formation. Studies have evaluated the modulation of αS amyloid formation by pH, chelators, ionic strength, and metals[9–12]. Mild changes affect amyloid conversion and the structural signature of disease-associated polymorphs. Accordingly, the nature of transient oligomers formed in the early stages of amyloid growth increases the difficulty in understanding the multitude of events behind amyloids. The heterogeneous aggregational nature of αS still challenges the amyloid community with open questions: Is there a general mechanism for amyloid formation? What are the cellular triggers for its conversion? How is a specific disease-related strain formed? Is there any relationship between the intermediate species and the end products?

Amyloid formation is a process involving interconverting protein species, and two theories exist to explain the early stages of growth: the nucleation-polymerization and the nucleation-conversion-polymerization model[13]. The first hypothesizes that αS monomers adopt a β-sheet structure that closely resembles the structure present in mature fibrils. Here, the rate-limiting step for the reaction to proceed is the fusion of small β-sheet multimers to form the minimum template capable of self-assembly. The second view assumes intramolecular and intermolecular hydrophobic collapse of monomeric αS that culminate in the formation of disordered oligomers that progressively evolve, at slow rates, into competent β-sheet oligomeric species. Several studies support the nucleation-conversion model[14,15]. Both theories predict a primary nucleation step in which soluble species are sequestered into oligomers of different sizes and β-sheet contents that progressively grow into protofibrils through monomer addition and thereby, mature amyloid filaments. Secondary processes, such as fibril fragmentation and surface-catalyzed nucleation, may also occur during the aggregation process[16]. The kinetics of amyloid formation is often visualized as a one-step process containing two major species (monomers and mature fibrils), and the interchangeable microscopic events of oligomer and protofibril formation are interpreted only by analytical solutions[17,18]. Thioflavin T (ThT) binding to oligomers and fibrils occurs with different quantum yields[19], but the distinction of these species during kinetic measurements has not been achieved so far.

The identification of different αS polymorphs[20–24] and the long-range variability within a mature filament represent obstacles to obtaining structural information about mature αS fibrils. However, near-atomic resolution maps of Tau amyloids and the Aβ peptide were reported using cryo-EM[25,26]. In αS, straight and twisted filaments support an in-register parallel β-sheet conformation[27], but buffer conditions can modulate their twisting[28]. By solid-state NMR, Tuttle and coauthors reported a single αS protofilament structure[29]. More recently, αS polymorphs

containing different quaternary arrangement and staggered protofilaments has been reported by cryo-EM[30–32]. To our knowledge, no comparison of frozen hydrated micrographs has been reported on αS filaments formed by genetic PD variants. However, structural approaches including small-angle X-ray scattering have uncovered some commonalities and structural differences of αS containing heritable mutations[33,34].

We performed a battery of biochemical, kinetic, and structural studies to understand the early stages of αS polymerization and the polymorphism of heritable αS variants. We systematically studied the kinetics of wildtype and three familial αS variants observed in early onset PD patients (i.e., A30P, E46K, and A53T) in response to ionic strengths. We provide evidence to distinguish oligomer and protofibril interconversions during ThT kinetic traces and a reconciliation of the nucleation-polymerization and nucleation-conversion-polymerization models to explain the different kinetic behaviors of wildtype αS and the A53T mutant. We provide a framework to study the early stages of amyloid conversion and place A53T with features that may help to explain the early onset of familial PD cases bearing this mutation.

## Results

**Ionic strength impacts αS kinetic traces.** We designed kinetic experiments with different concentrations of NaCl (Fig. 1a–d and Supplementary Fig. 1a–d) to be consistent with the physiological gradient of Na$^+$ observed across the membrane of neurons[35]. At the end of reactions (~90 h), we performed TEM of the end products (Fig. 1e) and absorbance measurements of the remaining soluble fractions (Fig. 1f–i). In addition, we observed the soluble species by TEM (Supplementary Fig. 2) and evaluated the final ThT fluorescence intensities (Fig. 1j–m). Because of the αS propensity to form higher order species, quality checks are provided in Supplementary Fig. 3.

The fluorescence of ThT during polymerization reactions revealed different behaviors among the studied variants. We observed increased slope values with increased ionic strengths (Supplementary Fig. 1e) suggesting that NaCl accelerates amyloid growth. When reactions occurred in the absence or presence of 1 and 10 mM salt, A30P and E46K showed a two-step transition involving three major ensemble of species (Fig. 1b, c). The corresponding soluble fractions tended to decrease with increased ionic strength (Fig. 1g, h). At 100 mM of salt, all variants exhibited a one-step transition and faster polymerization, as observed by decreased half-times (T$_{50\%}$, Supplementary Fig. 1f). The two major species at this situation are monomers at the beginning and mature fibrils as the final product (Fig. 1a–d, blue traces). Surprisingly, wildtype and A53T exhibited a three-step transition in the absence of salt, i.e., four major ensemble of species were discriminated (Fig. 1a, d; black traces). The multistep transitions were assessed for data reproducibility (Supplementary Fig. 4a, b). Compared to αS monomers (Fig. 1f, i; white bars), the soluble fractions of these variants showed an abrupt and monotonic decrease with increased ionic strength (Fig. 1f, i). The oligomers observed by TEM after kinetics most likely represent the soluble fractions of all end products and very few protofilaments decorated with oligomers were present (Supplementary Fig. 2). It is difficult to discriminate any contribution the soluble oligomers might have done to the formation of the fibrillar material. It is puzzling whether they might represent off-pathway material or had any participation on either slowing down the kinetic traces or accelerating it by serving as a source of competent monomeric species to growing fibrils.

TEM of the end products did not show appreciable differences in filament morphologies but instead revealed a distinction in filament length. The length of the filaments decreased with

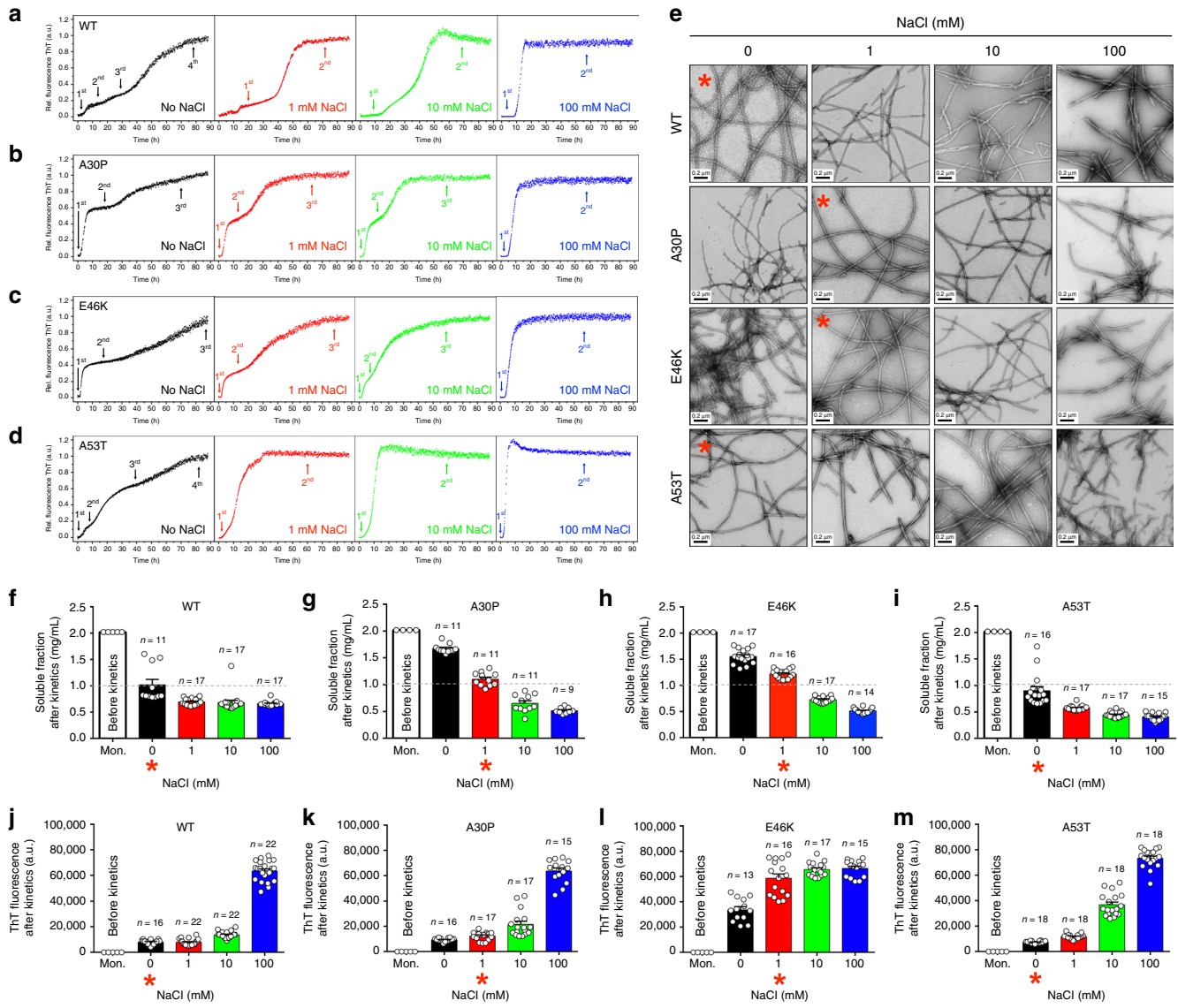

**Fig. 1** Modulation of αS aggregation by ionic strength. **a–d** Long-range kinetics of αS heritable variants at different sodium chloride concentrations monitored by the fluorescence of ThT. The curves are the average of several independent experiments (minimum of $n = 5$). Arrows containing 1st, 2nd, 3rd, and 4th indicate the detected stages of each kinetic trace. **e** Negatively stained TEM micrographs of mature αS fibrils obtained after ThT kinetic measurements (i.e., fibrillar fraction). Red asterisks represent the conditions selected for further analysis in which the amyloid conversion achieved equilibrium. **f–i** Scatter plots representing the soluble fractions ± s.e.m. and **j–m**, the ThT fluorescence counts ± s.e.m. after independent experiments. Mon. refers to αS monomers measured immediately before the start of the reaction. The asterisks in panels (**f–m**) mark the same conditions of panel (**e**). For sample size, check figure

increased ionic strength, most easily observed when comparing the no salt condition with the 100 mM condition for wildtype and A53T (Fig. 1e).

Asterisks in Fig. 1e represent the conditions selected for further analysis. These long amyloid fibrils are formed by two intertwined protofilaments, as already reported for other amyloids[25–27]. Although A30P and E46K filaments in the absence of salt were imaged (Fig. 1e), they were hardly detected during the searching procedure suggesting these fibrils represent a minor population. Further, a reasonable amount of the soluble fraction remained at the end of independent runs (Fig. 1g, h; black bars), suggesting that amyloid conversion is not favored under these conditions.

To use ThT fluorescence intensities as a quantitative measurement of general structural changes, we calibrate the linear range of its emission[36] (Supplementary Fig. 1g). The wildtype and A53T αS showed similar amounts of ThT binding in the absence of salt,

with means and standard errors of ~8000 ± 500 ($n = 16$) and ~7475 ± 245 ($n = 18$) counts, respectively (Fig. 1j, m; black bars). For A30P in the absence of salt (Fig. 1k; black bars), the ThT levels were likely similar to wildtype and A53T at the same condition, but the A30P soluble fraction showed a greater proportion when compared to the others (compare Fig. 1g; black bars with Fig. 1f, i; black bars). For E46K in the absence of salt, the soluble fraction is proportional to A30P (Fig. 1g, h; black bars), but ThT levels for E46K are at least twice (Fig. 1k, l; black bars). For wildtype and A53T in the absence of salt, the low ThT counts were mostly the same at 1 mM (Fig. 1j, m; red bars). ThT levels of A53T fibrils in the presence of 10 mM of salt were approximately twice when compared to wildtype (Fig. 1j, m; green bars) but exhibited similarly an abrupt increase in ThT counts at 100 mM salt (Fig. 1j, m; blue bars). In contrast, A30P and E46K under 1 mM salt showed different ThT binding with

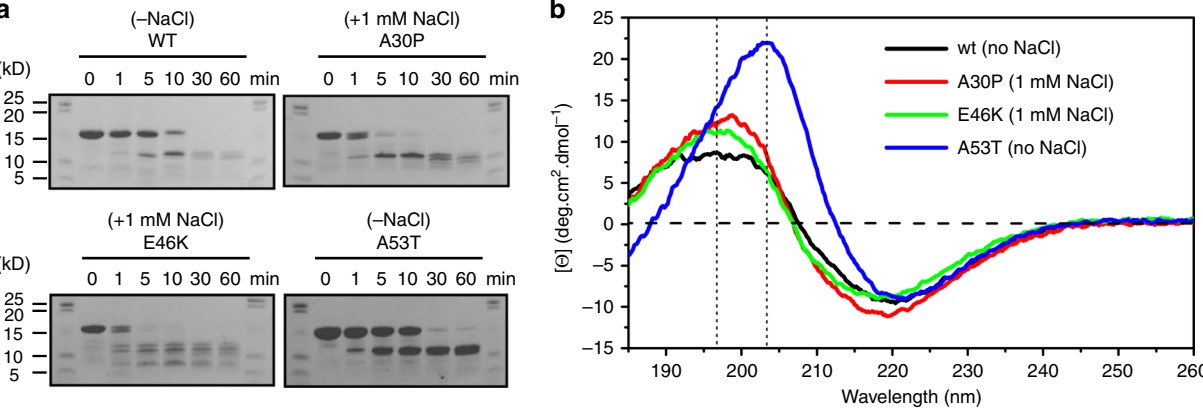

**Fig. 2** Mature fibrils of heritable αS variants are structurally different. **a** Proteinase k degradation profiles of selected fibrils in Fig. 1e (red asterisks) monitored over time on Coomassie blue-stained homogeneous SDS-PAGE gels. The lanes show increasing incubation times in min. **b** Far-UV circular dichroism spectra of αS fibrils. WT wildtype

means of ~12,380 ± 793 ($n = 17$) and 58,720 ± 3414 ($n = 16$) counts, respectively (Fig. 1k, l; red bars). The different values of ThT observed between wildtype/A53T (Fig. 1j–m; black bars), A30P (Fig. 1k; red bars), and E46K (Fig. 1l; red bars) are not attributed to a distinct amount of fibrillar material, as we measured similar concentrations of the soluble species (~1 mg mL$^{-1}$) at equilibrium for the selected conditions (Fig. 1f–i; asterisks). Other conditions in which we did not achieve similar soluble fraction measurements may have been affected either by the formation of oligomers that do not contribute for fibril elongation or amorphous aggregates that improperly diverted monomers from the amyloid growth.

Supporting the formation of structurally different filaments among αS variants (Fig. 1e; asterisks), the fragmentation profile from proteinase K (PK) digestion assays revealed similar bands between wildtype and A53T filaments (Fig. 2a). In contrast, different digestion patterns were observed between A30P and E46K filaments (Fig. 2a), confirming that wildtype and A53T filaments share conformational similarities, whereas A30P and E46K are structurally different. CD data of the fibrillar materials revealed the typical β-sheet pattern of amyloids containing negative bands at 218 nm and positive bands at 195 nm. The A53T fibrils revealed a dislocation of the positive band attributed to π–π* amide group electronic transitions from 195 to 200–205 nm (Fig. 2b).

**αS heritable variants form distinctly twisted filaments**. We performed cryo-EM to elucidate that αS fibrils containing heritable mutations (Fig. 1e, asterisks) are structurally different (Fig. 3). Unlike A30P and E46K, the wildtype and A53T exhibited similar morphologies based on a visual inspection of the frozen hydrated micrographs (Fig. 3a) and power spectrum evaluation of extracted filaments (Fig. 3b, c). The Fourier transformation of extracted filaments (power spectrum) allows the measurement of filaments pitch, i.e., the distance required to travel 360° around the filament axis. Dissimilarities of pitch distances among fibrils provide evidence of polymorphism at the quaternary structural level. As expected for a cross-β architecture, we observed the invariant 1/4.7 Å$^{-1}$ spacing between αS chains along the fibril axis (Fig. 3b). The power spectra of A30P and E46K filaments revealed additional low-resolution layer lines close to the equator corresponding to ~1/398 Å$^{-1}$ and ~1/1460 Å$^{-1}$, respectively (Fig. 3c and insets). It directly informs half of the pitch value of studied fibrils. For wildtype and A53T, no further layer lines were observed close to the equator (Fig. 3c), supporting the long pitch periodicity of these filaments (Fig. 3a). The frozen hydrated

micrographs of wildtype and A53T exhibited similar morphologies either in the absence (Fig. 3a) or presence of 1 mM salt (Supplementary Fig. 5a, b). The findings confirm that unlike A30P and E46K, the wildtype and A53T filaments share quaternary structural similarities regardless of subtle changes in ionic strength (up to 1 mM NaCl).

**Ionic strength impacts αS fibrils length and conformation**. To support that increasing ionic strength decreases fibril length, we measure the length distribution of filaments grown in the absence and presence of 100 mM salt (Fig. 4a–f). The presence of shorter filaments with increasing ionic strength (Fig. 4c, d), together with increased slope values (Supplementary Fig. 1e), shorter lag phases and accelerated kinetics with higher salt (Fig. 1a–d and Supplementary Fig. 1f), supports the impact of salt on αS growth kinetics. It is hard to know whether fibril fragmentation may also take place and might contribute on αS growth, but the morphology of some shorter fibrils are likely consistent to broken mature filaments (Fig. 4f).

To study the effects of ionic strength on αS filament structure, we performed PK digestion for wildtype and A53T filaments formed in the absence or presence of increasing salt (Fig. 5a–d and Supplementary Fig. 6a, b). We observed a distinct profile of PK-digested bands when comparing wildtype and A53T filaments grown in the absence of salt to those formed in 100 mM of salt (Fig. 5a, c, and Supplementary Fig. 6). It suggests that salt may cause conformational differences in αS filaments. To confirm this, we designed a controlled PK digestion coupled with electrospray ionization mass spectrometry (ESI-MS) to assess the peptide signature of wildtype and A53T filaments in the absence and presence of salt. We detected peptides up to 4 kD and found sequence coverage higher than 80% in all runs (Supplementary Fig. 7). The region from residues 20 to 30 was never observed, presumably because it is easily digested (Supplementary Fig. 7a–d). The most abundant species were from the C-terminal, comprising residues 90 to 140, and these species may represent peptides that are harder to digest. We show the prominent cuts (Supplementary Fig. 7), which mostly occurred at L113 and Y125. The Venn diagrams show a comparison of the unique set of identified peptides when wildtype and A53T filaments are formed in the absence and presence of salt (Fig. 5e, f). Mapping these unique peptides and their abundance by αS sequence revealed a distinct cleavage pattern, confirming that conformational differences arise when wildtype and A53T filaments are formed under the influence of NaCl.

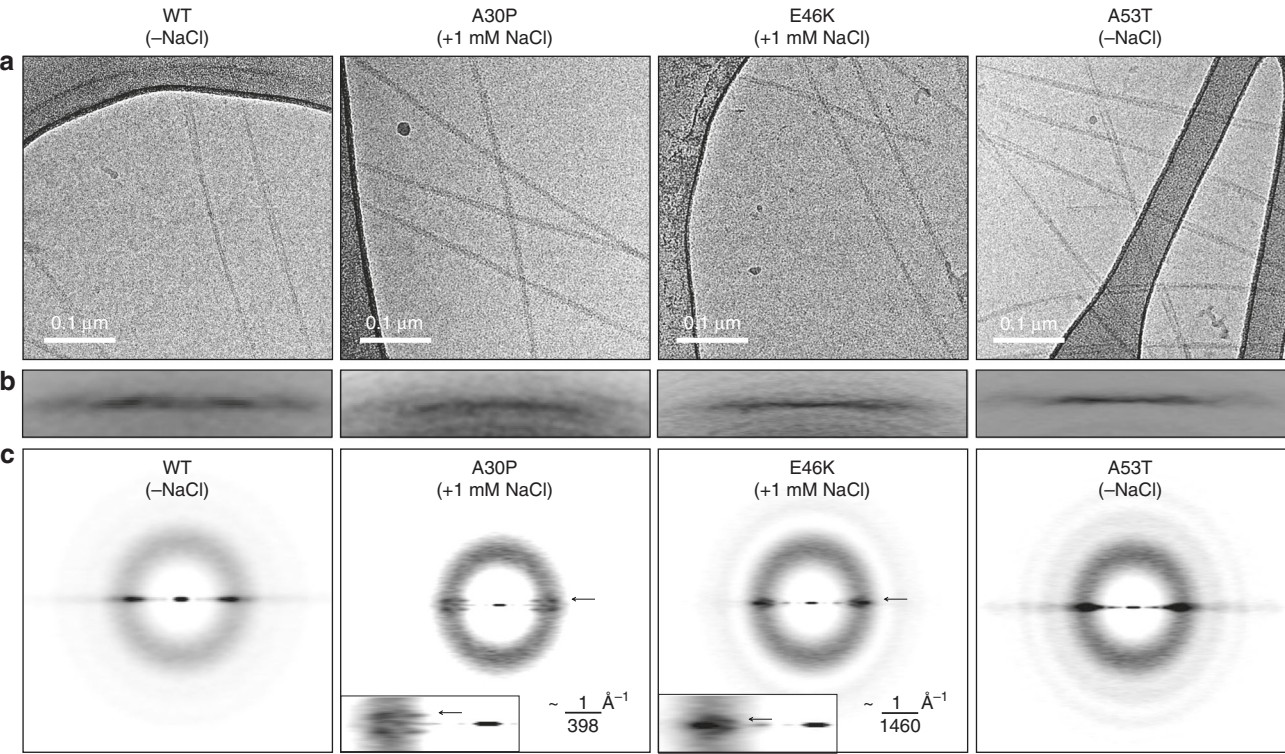

**Fig. 3** Structural properties of heritable αS mature fibrils. **a** Representative frozen hydrated micrographs of αS fibrils (wildtype (WT), A30P, E46K, and A53T) under the conditions selected in this study. **b** The power spectrum segment is showing the typical $1/4.7$ Å$^{-1}$ high-resolution layer line of the studied amyloid fibrils. **c** The power spectrum region is highlighting the additional low-resolution layer lines (arrows and inset) observed for the A30P (~$1/398$ Å$^{-1}$) and E46K (~$1/1460$ Å$^{-1}$) fibrils. The WT and A53T fibrils exhibited no additional layer lines in this region

Hereafter, we turned our attention to the kinetic traces of wildtype and A53T as an attempt to understand how A53T behave in early onset cases of familiar PD.

**A53T elongates seeds and overcome surface needs for growth.**
To understand the effects of salt on αS amyloid formation, we evaluated the quiescent ThT kinetics of wildtype and A53T in the absence or presence of 100 mM salt and no seeds. The soluble fractions had similar amounts of soluble species as those measured for the initial monomers (Fig. 6a). It suggests that under the studied timeframe (~90 h), polymerization of mature αS fibrils rarely occurs and longer incubation periods would be necessary. Evaluation of ThT fluorescence showed no binding for wildtype and A53T in the absence of salt, but a lower ThT signal for wildtype and a greater increment for A53T in the presence of salt (Fig. 6b). We observed that under quiescent conditions at neutral pH and using preformed seeds (Supplementary Fig. 8a), A53T forms amyloid fibrils (Fig. 6c). It indicates that secondary nucleation might happen for this mutant at neutral pH, but because A53T does not substantially modify the charge state of αS, we became conservative to exclusively attribute secondary nucleation to the observed kinetic traces at neutral pH. Sedimentation would also explain the observed effects. Notwithstanding, it is interesting because αS was shown to aggregate through secondary nucleation only at mildly acidic conditions[11], revealing a potential advantage of this mutant at physiological pH.

The dominant secondary process of αS aggregation occurs via nucleation of monomers on the fibril surface instead of fibril fragmentation[37]. To evaluate the influence of surfaces on αS aggregation, we performed wildtype and A53T kinetics using standard polypropylene (PPP) tubes and low adhesion PPP tubes

followed by quantification of the soluble and fibrillar fractions (Fig. 6d). Unlike wildtype, the A53T was able to elongate fibrils even when incubated in a low adhesion surface for growth (Fig. 6d, arrow). By using TEM images (Fig. 6e), we confirmed the presence of A53T fibrils suggesting that A53T overcomes the wildtype αS surface requirement for growth.

Increasing the mass of seeds under shaking conditions, we detected distinctly different behaviors when comparing wildtype and A53T with and without 100 mM salt (Fig. 7a–h). The multistep transition observed was gradually suppressed when the amount of seeds increased (Fig. 7a, c). The exponential burst and complete suppression of the lag phase were observed after using higher amounts of seeds (Fig. 7a, c). Soluble and fibrillar fractions in the absence of salt revealed the consumption of soluble species and the formation of more fibrils with increasing seeds in similar manners for wildtype and A53T (Fig. 7e, g; descending and ascending black bars). Although fragmentation is present due to shaking, the seeding reactions suggest that secondary nucleation events may have a role on αS growth kinetics.

The properties of the seeding reactions and their soluble and fibrillar profiles drastically changed in the presence of 100 mM salt (Fig. 7b, d). The wildtype soluble fraction increased with increasing amounts of seeds (Fig. 7f; ascending black bar), but the reactions revealed no substantial formation of new fibrils with increasing seed quantity (Fig. 7f, invariant black bar). Remarkably, the fibril mass of the A53T mutant easily increases with increasing amounts of seeds (Fig. 7h, ascending black bar). The wildtype soluble fractions increasing (Fig. 7f) reveal that either the added monomers or the seeds were not consumed in the reaction. Control experiments revealed that a solution composed exclusively of seeds contained similar amounts of species in the soluble and fibrillar fractions (Supplementary Fig. 8b, c). Nevertheless, the spike-like morphology of the soluble portions of the seed solution differed most from the

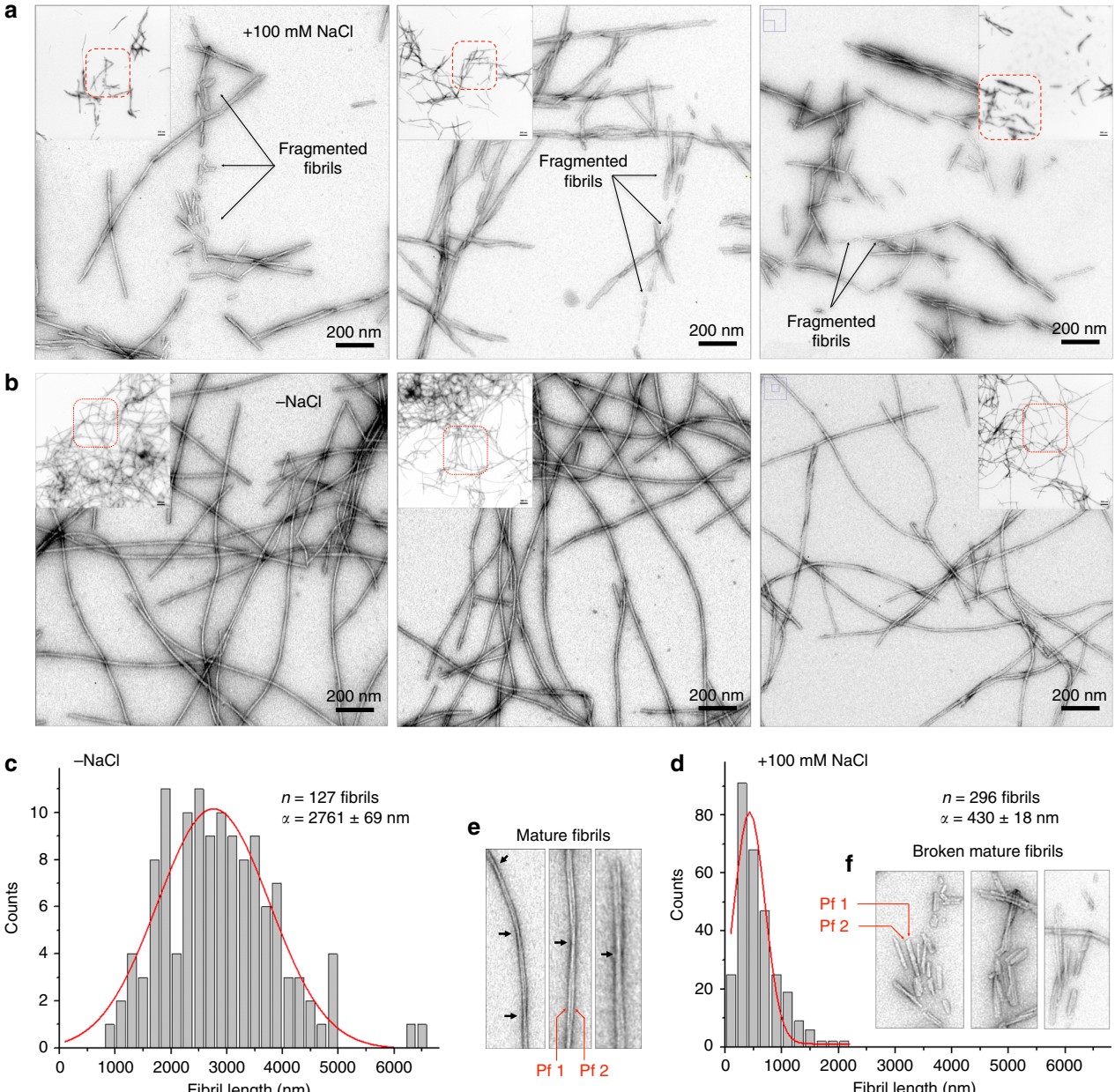

**Fig. 4** Ionic strength drastically influences the length distribution of αS filaments. **a**, **b** Negative staining micrographs of wildtype αS filaments grown (**a**) in the presence of 100 mM of NaCl and (**b**) in the absence of salt. Insets show low magnified micrographs and dashed red lines the zoomed regions. **c**, **d** Frequency of length distribution for evaluated numbers of filaments (n) and the average length (α) obtained in the absence of salt (**c**) and in the presence of 100 mM of NaCl (**d**). **e**, **f** Zoomed filaments to better represent the effects of high ionic strength during the αS growth kinetics. Black arrows show filament crossovers. Pf stands for protofilament

oligomeric morphology obtained from the soluble fraction after experiments evaluating the wildtype kinetics in which seeds were added (Fig. 7f and Supplementary Fig. 8d). It supports that the increased amount of wildtype soluble species observed in Fig. 7f most likely represent the formation of oligomers from added monomers and not the accumulation of added seeds. Collectively, the data identifies features that may help to explain the early onset illness of PD cases bearing the heritable A53T mutation. In the presence of increasing amounts of seeds, unlike wildtype, A53T is able to increase the fibril mass in the presence of physiological relevant ionic strength (Fig. 7f, h). Further, A53T forms amyloid fibrils via secondary nucleation processes at neutral pH under quiescent conditions (Fig. 6c) and overcome wildtype surface attributes for growth (Fig. 6d, e).

**A53T rapidly nucleate competent species for amyloid growth.** To better understand the multistep kinetic behavior captured for wildtype and A53T, we designed the procedure described in the Fig. 8a. During a ThT kinetic run, aliquots were taken at the half-time of each observed transition (Fig. 8b) and subjected to ultracentrifugation. The supernatant fractions were then immediately assessed by A-Sec (Fig. 8c, d) and prepared for TEM (Fig. 8b), and the pellets were observed by TEM (Fig. 8b and Supplementary Fig. 9). Although αS monomers represented the predominant species during temporal A-Sec (Fig. 8c, d; labeled M), a minor but clear separation was found between large and small oligomers (insets Fig. 8c, d; labeled LgO and SmO, respectively). Immediately before the kinetic experiment, A-Sec of wildtype and A53T monomers revealed a steady-state equilibrium

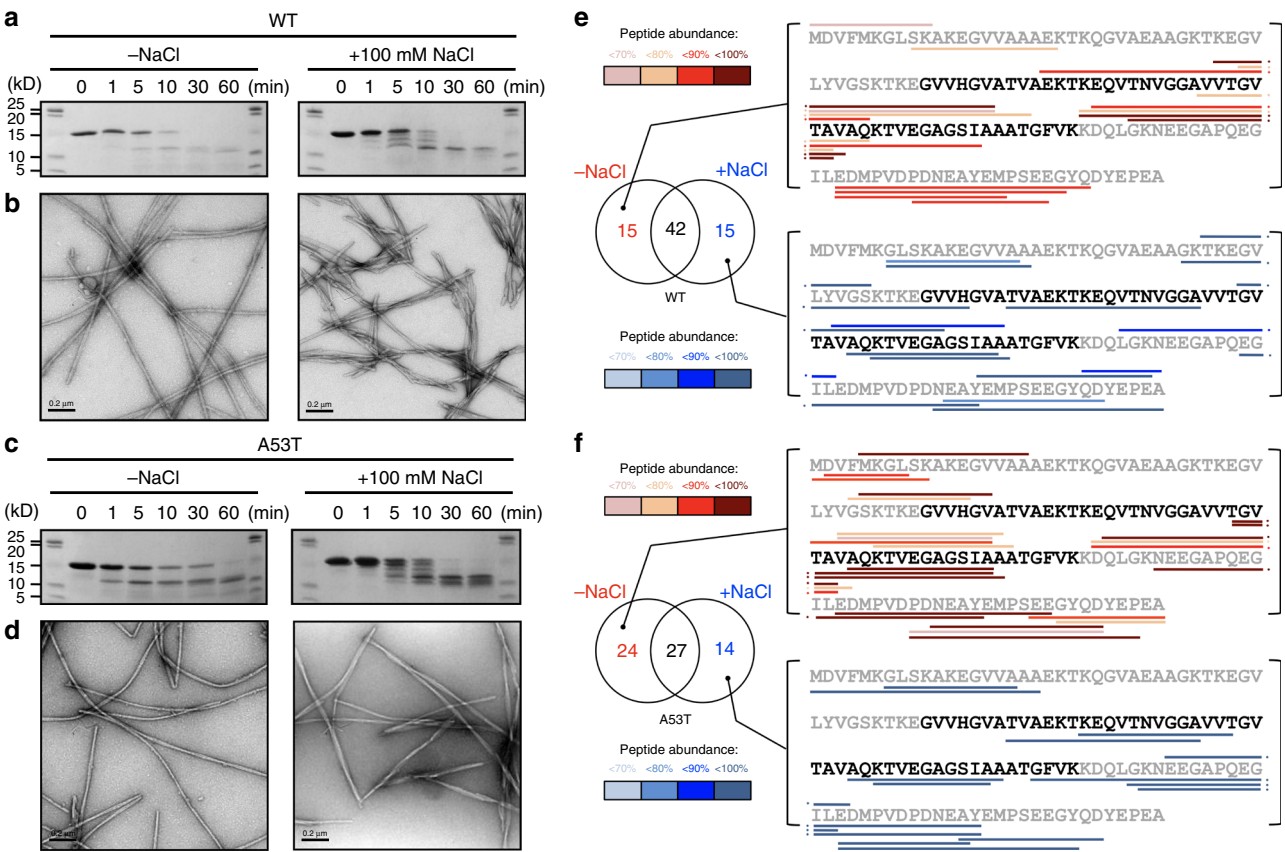

**Fig. 5** Conformational diversities modulated by sodium chloride. **a**, **c** Wildtype (WT) and A53T fibrils produced in the absence or presence of 100 mM salt were evaluated via proteinase K digestion assays and monitored over time on Coomassie blue-stained homogeneous SDS-PAGE gels. **b**, **d** Representative micrographs of negatively stained (**b**) WT and (**d**) A53T fibrils grown in the absence or presence of salt. **e**, **f** The Venn diagrams show the number of unique peptides identified for the WT and A53T fibrils produced in the absence (−NaCl) or presence of 100 mM salt (+NaCl). The horizontal lines in the αS primary sequence highlight the unique peptides identified by mass spectrometry runs. The color scheme shows the peptide abundance for each studied condition. The colored lines with dots refer to the same peptide. The amino acids in black correspond to the predicted hydrophobic core of αS fibrils

with a minor population of SmO (Fig. 8c, d; T0). While the amount of SmO decreased and the amount of LgO increased throughout the wildtype kinetic experiment, A53T exhibited no appreciable changes in SmO, and LgO was not detected (insets Fig. 8c, d). Instead, temporal A-Sec runs for A53T revealed a prominent peak at ~11 mL, which was not resolved for wildtype (Fig. 8d; multimers (Mts)). The disordered nature of αS monomers explains elution in the Superose 6 with an estimated weight of 68 kD. The prominent peak observed for A53T corresponds to species of ~89 kD and most likely represent A53T conformational multimers (Fig. 8d; Mts). The soluble fractions evaluated by TEM after 6 h of kinetics confirmed an enriched population of pleomorphic wildtype oligomers with different sizes. We defined based on a naked eye inspection, at least three different oligomeric species and classified them according to their sizes and staining tendencies. SmOs were dominant and presented a white color appearance. Growing oligomers (GrOs) were intermediate in size, and LgOs were more negatively stained (Fig. 8e and Supplementary Fig. 9a, b). The remaining LgOs were observed by TEM after 46 h of wildtype kinetics (Fig. 8b), and were in line with the increased LgO peak observed by A-Sec (inset of Fig. 8c; T46h). Because the soluble aliquots at each time were divided for A-Sec and TEM, we attribute the LgOs and SmOs separated by A-Sec runs to the same LgOs and SmOs characterized by TEM. The results suggest that some LgOs formed by wildtype αS in the absence of salt accumulate during the kinetic reaction, while some competent SmOs are most likely consumed to initiate the fibrillar pool. For A53T, the lack of LgO and the steady-state equilibrium

among SmOs and smaller multimers may satisfy the requirements of a continuous pool of interconverting species capable of forming the fibrillar fraction rapidly. After 10 and 20 h of kinetic runs, the fibrillar fractions showed growing protofibrils, and after 46 h, we mostly observed mature filaments (Fig. 8b and Supplementary Fig. 9c; pellet).

We provide evidence to argue that the differences observed between wildtype and A53T kinetics arise from distinctly different intermediate species formed in the early stages of aggregation (Figs. 8 and 9). This dissimilar behavior has no impact on the end products but does affect the growth rates, as wildtype and A53T mature filaments formed in the absence of salt are indistinguishable based on the ThT fluorescence counts (Fig. 1j, m; black bars), PK digestion profiles (Fig. 2a), and Fourier analysis (Fig. 3 and Supplementary Fig. 5). At higher salt concentrations, instead, conformational diversity arises (Fig. 5), elongation occurs faster (Fig. 1a, d; black and blue traces), and fibrils become shorter (Fig. 4a–f) but, unlike wildtype, the A53T continuously increase the fibril mass in the presence of preformed seeds (Fig. 7f, h) and overcome surface attributes for growth (Fig. 6d, e).

## Discussion

Amyloid formation is crucial to explain the factors underlying the formation of toxic oligomers and fibrils and their role in neurodegeneration. We expand our knowledge to distinguish oligomer and protofibril interconversions via the readout of ThT kinetics

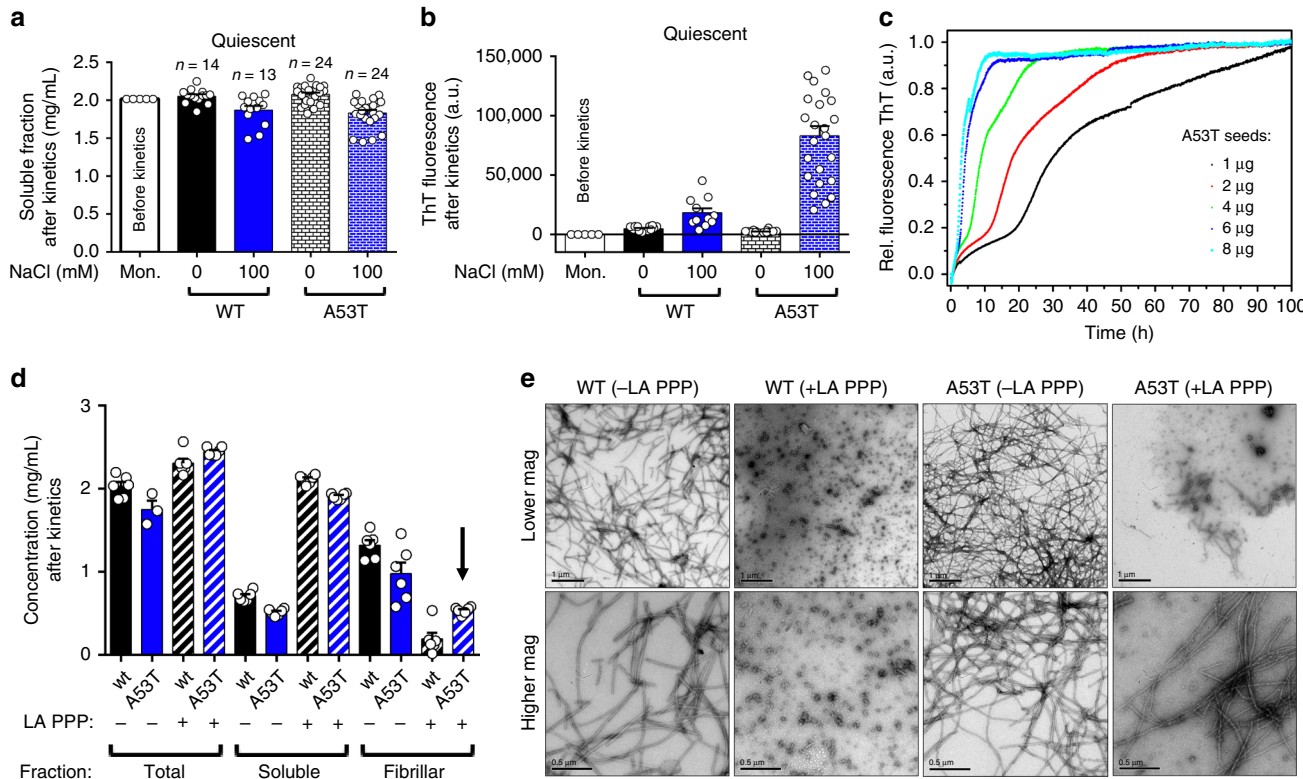

**Fig. 6** Secondary nucleation processes of A53T growth kinetics. **a** Scatter plots representing the soluble fractions and **b** ThT counts of wildtype (WT) and A53T αS after ThT kinetic measurement performed under quiescent conditions in the absence and presence of 100 mM salt. **c** ThT growth kinetics of A53T (70 μM) in the presence of increasing concentrations of seeds, low salt concentration (1 mM NaCl) and neutral pH under quiescent conditions. **d** Scatter plots of total, soluble, and fibrillar fractions of WT and A53T grown for 6 days at 37 °C, 600 rpm in the presence of 1 mM of salt in standard polypropylene and low adhesion polypropylene tubes (LA PPP). Arrow shows the fibrillar fraction of A53T grown in LA PPP. **e** Negative staining micrographs of WT and A53T grown in standard (−LA PPP) or low adhesion propylene tubes (+LA PPP). For sample size, check figure

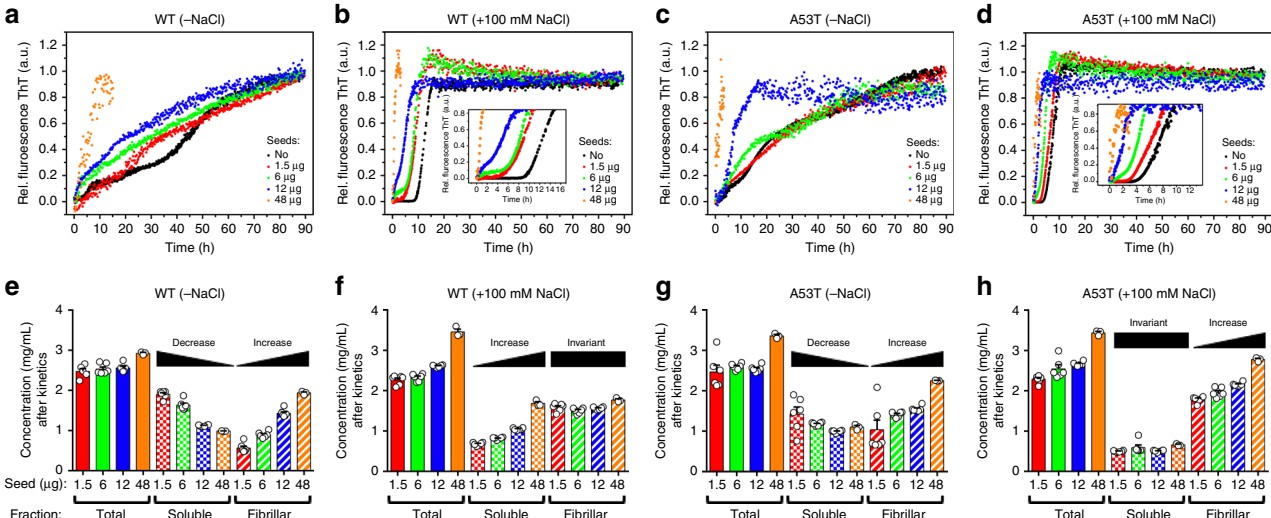

**Fig. 7** Dissimilar behaviors of wildtype and A53T αS growth kinetics. **a–d** Long-term ThT kinetic measurements of wildtype (WT) and A53T αS with increasing amounts of seeds (in μg) performed in the absence or presence of 100 mM salt under agitation. **e–h** Scatter plots representing the total, soluble, and the corresponding fibrillar fractions as a function of increasing amounts of seeds for WT and A53T αS under the studied conditions. Averages ± s.e.m. are shown for at least three (n > 3) independent experiments

and provide a hypothesis reconciling the nucleation-polymerization and nucleation-conversion-polymerization models to explain the dissimilar kinetic behaviors of wildtype αS and the A53T mutant.

αS is dependent on a heterogeneous nucleation process at interfaces including the polymer-water interface[37,38], the air-water interface[39], or lipid-water interface[40]. We evidenced that unlike wildtype, A53T elongates fibrils when incubated in a low

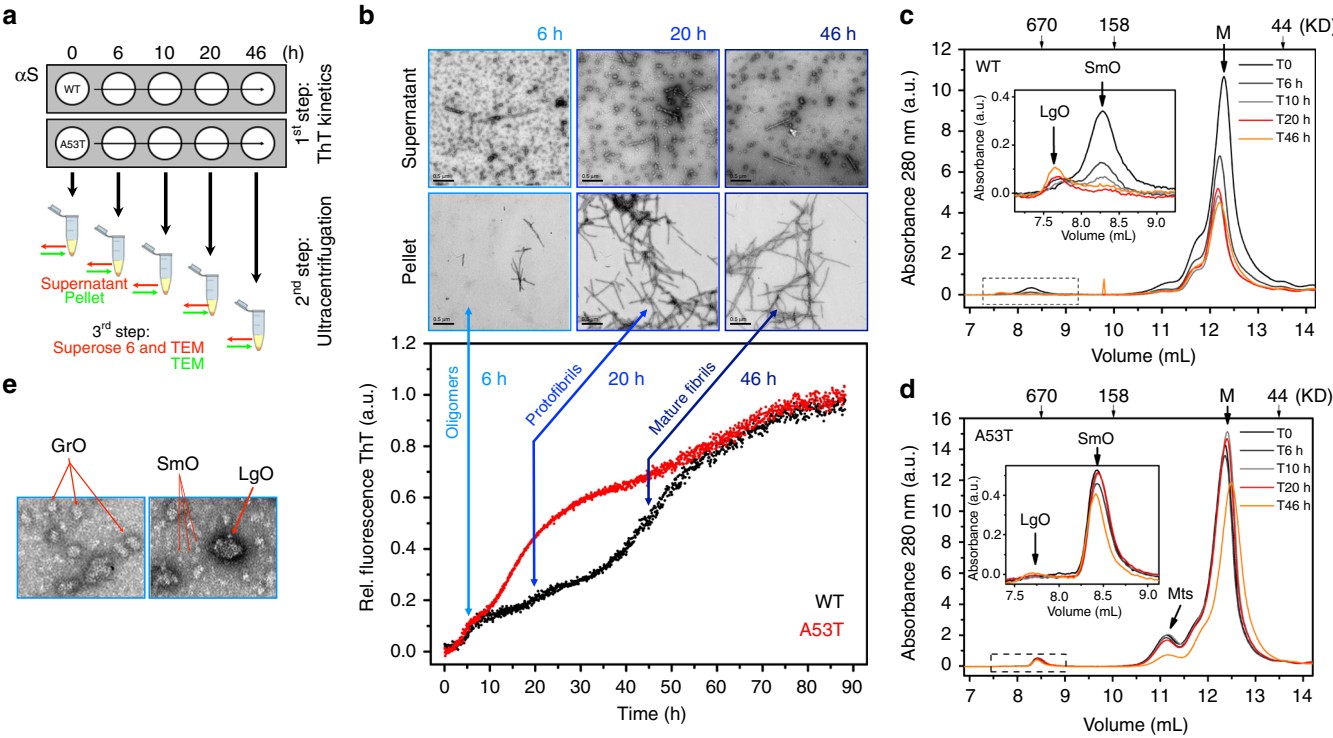

**Fig. 8 Kinetic dissimilarities observed between wildtype and A53T αS. a** The schematic shows the experimental approach. During the first step of ThT kinetic measurements (−NaCl), aliquots were taken at different times and subjected to ultracentrifugation (second step). The supernatants were then assessed at various times by analytical size exclusion chromatography (third step), and the supernatants and pellets were imaged by negative staining TEM (fourth step). **b** Aliquots taken at approximately the half-time (6, 20, and 46 h) of each transition observed during the wildtype (WT) and A53T ThT kinetic measurements (−NaCl). Negatively stained micrographs of WT αS showing representative soluble and pellet fractions over time. **c, d** Analytical size exclusion chromatography profiles of (**c**) WT and (**d**) A53T αS over time (0, 6, 10, 20, and 46 h). The elution volumes corresponding to the oligomeric fractions are highlighted with dashed lines and magnified in the insets. M Monomers, LgO Large oligomers, SmO Small oligomers, Mts multimers. **e** Zoomed-in regions of the soluble fractions after 6 h of ThT kinetic measurements. Oligomers of different sizes are highlighted. SmO; small oligomers, GrO; growing oligomers, and LgO; large oligomers

adhesion surface. It indicates that A53T has the ability to overcome the polymer-water interface requirement of the wildtype. It is true that A53T may compensate this by accelerating the primary nucleation at the air-water interface, but this is awaiting further exploration.

Amyloid conversion is frequently obtained from time course measurements of fluorimetric probes, but interpretations are limited to parameters directly extracted from the sigmoidal behavior[41]. The application of master equations instead has provided support to the understanding of the microscopic events and rate constants of amyloid growth and fragmentation[17,18]. Although the ThT fluorescence increases upon binding to cross-β sheets and amyloid progresses through increases in the β-sheet content of oligomers, protofibrils, and mature filaments, the contributions of the microscopic events of oligomer accumulation and protofibril formation to the macroscopic-level of kinetic measurements were not assessed thus far. These nucleating species may either interconvert rapidly or may not have formed in sufficient amounts for detection by ThT. Alternatively, because amyloid formation is sensitive to changes in environmental conditions[9,11], it is possible that instead of primary nucleation events, secondary events would dominate the kinetic course and prevent the formation of sufficient oligomers and protofibrils for detection by ThT. We solved this problem by performing systematic time evolution experiments with various concentrations of NaCl. We identified a highly reproducible multistep transition for wildtype and A53T amyloid conversion in the absence of salt

(Fig. 1a–d, and Supplementary Fig. 4) and characterized each stage utilizing A-Sec (Fig. 8c, d) and TEM (Fig. 8b and Supplementary Fig. 9). Secondary nucleation was shown to dominate αS aggregation via nucleation of monomers on the fibril surface instead of fibril fragmentation[37]. We believe that at 100 mM of salt and under agitation, fragmentation may also contribute to the αS growth kinetics. The NaCl-induced conformational diversity observed for wildtype and A53T fibrils (Fig. 5) is supported by infrared data showing a shift from parallel to anti-parallel β-sheets at different salt concentrations[42]. The far-UV spectrum of A53T fibrils grown in the absence of salt revealed a dislocated π–π* electronic transition consistent with the presence of parallel β-sheets[43]. Due to the H-bond organization of parallel β-sheets, this secondary architecture is less stable than the anti-parallel sheet organization. This may allow to A53T monomers a faster screening for competent converting species at the early stages of amyloid nucleation.

Hydration and electrostatic interactions dictates αS aggregation[9]. When fragmented fibrils are present, new fibril ends will be available for monomer incorporation catalyzing the elongation step. Accordingly, the elongation rate of amyloid fibrils increases with ionic strength[10]. This observation may explain the exponential bursts (Fig. 1a–d) and the production of shorter mature fibrils at higher salt concentration (Fig. 4a–f), as new competing ends will be available for growth. In addition, the increased ThT counts observed at higher salt concentration (Fig. 1j–m; blue bars) might be a consequence of the conformational differences

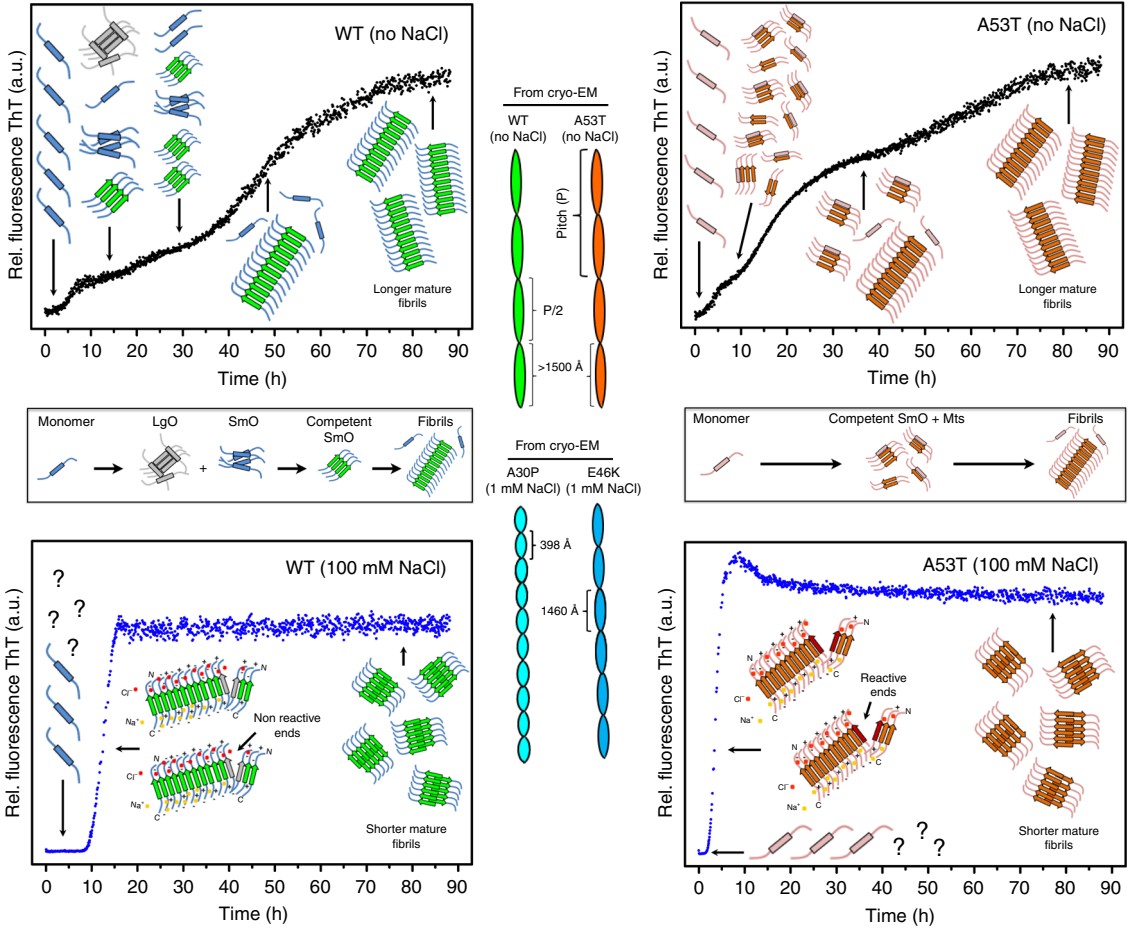

**Fig. 9** The dissimilar behaviors captured by ThT kinetic measurements of wildtype (WT) and A53T aggregation under low and physiological relevant ionic strengths may be explained by the different pattern of oligomeric species at the early stages of aggregation, a condition that may provide advantages during the growth kinetics of this mutant. The quaternary structural polymorphism observed by cryo-EM of αS heritable variants is shown by distinctly different filament twists. The question character (?) stands for species there are not captured by ThT kinetics in the presence of higher ionic strength

arising between fibrils (Fig. 5a–f) and favoring ThT steric accommodation or ThT binding to the increased amount of reactive fibril ends.

Conceivably, the effects $Na^+$ and $Cl^-$ may have on αS to elongate fibrils would be a combination of events, including their direct binding and the hydration disturbance surrounding initial filaments; together with shear forces due to agitation, it might favor fibril breakage to some extent. In support of the role water molecules may have in fibril stability, when glycerol preferentially excludes water from the regions around fibrils, the fibrils are stabilized against hydrostatic pressure-induced breakage[44]. Accordingly, the presence of exposed charges at the N-termini and C-termini of αS would favor direct binding of $Na^+$ and $Cl^-$ and electrostatic screening to initiate a competent nuclei for growth, a condition that may increase the elongation rate of growing filaments.

Reports have established the relationship between fibril length and their cytotoxic effects[45,46]. For instance, shorter fibrils formed by αS, β2-microglobulin, and lysozyme are more cytotoxic[45]. More remarkably, there is a relation between the size of αS fibrils and their ability to act as seeds for the prion-like conversion of the normal αS into fibrils[46]. Notably, the α3 subunit of $Na^+/K^+$ ATPase (α3-NKA) was identified as a surface partner of αS fibrils[47] compromising the removal of $Na^+$ from the intracellular space. In resting neurons, intracellular concentration of $Na^+$ would provide a favorable scenario to keep long αS fibrils within the cell. During neuronal activity, $Na^+$ fluctuations might favor

electrostatic screening what would accelerate αS primary nucleation and the fibril growth rate generating shorter fibrils. Shorter segments are more likely to be extruded, altering the host cell membrane interactions and intracellular signaling which may provide cell-to-cell transmission and propagation effects to guarantee disease and spreading[3,45,46].

We detected oligomer and protofibril interconversions during ThT kinetics (Fig. 8b). This is critical for characterizing the intermediate species participating in amyloid growth and how different αS variants behave. Oligomers are the primary cause of the toxicity associated with neurodegeneration[48], and several mechanisms were proposed to explain the toxic effects[49–52], including membrane perturbation[53,54]. Although the A53T dominates the growth rates during αS aggregation[55] and single-molecule FRET showed possible structural differences of oligomeric ensembles in heritable variants[56], there are no mechanistic insights to explain the dissimilar kinetic behaviors of wildtype αS and the A53T variant. By modulating salt concentrations of polymerization reactions, we pinpoint key determinants of the growth rate disparities between wildtype and A53T αS. First, wildtype LgOs accumulate, while A53T LgOs are not detected. Second, wildtype SmOs are progressively consumed by the fibrillar pool, while A53T SmO levels are just marginally changed. Finally, a prominent peak most likely representing small conformational multimers was detected for A53T but was negligible for wildtype. Importantly, these differences were observed during ThT kinetic traces in which equilibrium was reached and amyloid

fibrils were identified as end products. Further, the initial αS monomeric species were not subjected to lyophilization or supercritical concentrations, as is often reported in protocols used to characterize oligomers[57,58]. We cannot determine whether the αS oligomers observed during our measurements are the same as the previously characterized type-A/-A* and type-B/-B* oligomers as small changes in oligomer preparation, such as lyophilization or the addition of compounds, may represent a source of heterogeneity.

We propose a hypothesis that reconciles the current models of amyloid growth: the nucleation-polymerization and nucleation-conversion-polymerization models. We stress that amyloid formation is not a process in which species are formed sequentially. Rather, population-weighted averages of multiple species coexist at each ThT stage in different proportions. For wildtype αS, the longer lag phases and the consumption of SmOs for integration into the fibrillar pool most likely correspond to a nucleation-conversion-polymerization model, in which disordered oligomers are first converted to more organized β-sheet structures and then start to form mature fibrils (Fig. 9). This process may explain the longer wildtype transitions in our ThT kinetics (Figs. 1a, d, 8b and Supplementary Fig. 4). Unlike wildtype, A53T had a constant pool of interconverting multimers and SmOs that better fits the nucleation-polymerization model, in which "ready-to-build" species are rapidly formed in the early stages of αS aggregation and are continuously accessible to incorporate αS monomers and guarantee the exponential burst during protofibril growth (Fig. 8b–d). The sequestration of wildtype monomers as LgOs probably limits the availability of conformational monomers that can be incorporated into growing fibrils. Because no substantial loss to LgOs occurs for A53T, this does not become a limiting step for A53T kinetic growth. A recent study showed that αS is prone to covalent dimerization triggered by dityrosine cross-links[59]. However it is still unknown whether the observed A53T multimers represent these kinds of αS covalent dimers. Thus far, by showing that A53T rapidly nucleates competent species for growth, continuously elongates fibrils in the presence of increasing amounts of preformed seeds and overcome wildtype surface attributes for growth, our findings place A53T with features to explain the early onset of familial PD cases bearing this mutation.

## Methods

**Preparation of monomeric αS**. Recombinant monomers of αS were grown based on the previously described protocol[60]. Protein purification was performed using the osmotic shock protocol with minor modifications for the dialysis step and protein storage. After αS precipitation with 50% w/v $(NH_4)_2SO_4$ and centrifugation $(25,000 \times g$ for 15 min, 4 °C), the pellet was resuspended in ca. 8–10 mL of 20 mM Tris-Cl (pH 8.0) containing 1 mM EDTA and dialyzed overnight (ca. 16 h) against 5 L of the same buffer using SnakeSkin dialysis tubing, 7 kD MWCO (Thermo Scientific, #68700). Dialyzed protein batches were filtered using 0.22 μm PVDF filters (Millipore, #SLGV033NS), directly frozen at −80 °C, and used for no more than 6 months. All kinetic traces were performed with fresh samples, ideally within 1–4 weeks after purification as reproducibility was hardly achieved with samples stored for longer periods (2–6 months) probably due to the accumulation of abnormal oligomeric species.

For kinetic and all further experiments in this study, aliquots of 500–700 μL (ca. 200–400 μM) were thawed on ice and further dialyzed (ca. 16 h) against 5 L of the following buffers: 5 mM Tris-Cl (pH 7.4) (−NaCl condition) or 10 mM Tris-Cl (pH 7.4) containing 1 mM EDTA and the corresponding NaCl concentration (1, 10, or 100 mM). Because of a buffer concentration of 10 mM, the 1 mM NaCl condition has a small effect on ionic strength. For this step, we used a Slide-A-Lyzer mini dialysis device, 10 kD MWCO (Thermo Scientific, #88401). The studied αS variants were dialyzed at the same time using the same 5 L buffer container for a chosen condition to exclude any variability in buffer composition among the studied constructs and increase the data robustness. A-Sec (250 μL per injection at a flow rate of 0.7 mL min⁻¹) using a Superdex 200 Increase 10/300 GL column (GE Lifesciences, #28-9909-44) was performed immediately after dialysis to check the purity of monomeric αS (Supplementary Fig. 3).

**Kinetic experiments**. After dialysis, the concentration of monomeric αS variants was estimated by the absorbance at 280 nm using a molar extinction coefficient of 5960 M⁻¹ cm⁻¹ and adjusted to 140 μM using the same dialyzed buffer for the chosen condition. We used clear-bottom 96-well plates (Thermo Scientific, #265301) with a final reaction volume of 100 μL in each well. We used at least two different protein batches. To exclude experimental variations as much as possible, we designed kinetic runs to ensure that all studied αS variants and buffer conditions were assessed in the same 96-well plate; thus, the observed effects were exclusively related to the specific αS construct and buffer composition. We used F1-ClipTip multichannel pipettes (ThermoScientific, #4661110 N, #4661130 N) and low retention ClipTip pipette tips (Thermo Scientific, #94410310, #94410210) to prepare the plates.

αS polymerization reactions were performed using 140 μM (~2 mg mL⁻¹) αS monomers and monitored by the fluorescence of 8 μM ThT (Sigma, #T3516). The ThT concentration used in our kinetic traces is within the linear response of the dye, thus it is a reliable metric to infer structural variations (Supplementary Fig. 1g). ThT emission was measured at 4 min intervals for ~90 h at $\lambda_{max} = 477$ nm upon excitation at 450 nm. To increase the data reproducibility among different wells, we used one 1/8" diameter Teflon ball per well (Polysciences, #17649). The plates were sealed with clear polyolefin sealing tape (Thermo Scientific, #232702) and loaded into a 37 °C prewarmed Spectra Max Gemini EM plate reader (Molecular Devices). The temperature was 37 °C throughout the entire kinetic measurement, and the orbital agitation was set to 10 s between reads unless stated otherwise. The photomultiplier gain was set to low (i.e., 6 flashes per reading), and the reads were set to be from the bottom of the plate. Low adhesion polypropylene tubes used for surface experiments were obtained from USA Scientific (#1415-2600).

The ThT fluorescence was scaled to the plateau level of fluorescence at the beginning and end of the kinetic traces to make the ionic strength effects on ThT transitions comparable (Fig. 1a–d). We used the last transition to plot the slope changes of all kinetic traces because it represents the greatest gain in ThT fluorescence, most likely explained by the increasing of the fibril mass (Supplementary Fig. 1a–d).

Outliers from the raw data in Supplementary Fig. 4 and from all other kinetic traces were excluded during averaging of five independent traces. Data exclusion was performed using an automated cutoff procedure in Wolfram Mathematica with no impact to the kinetic tendency behavior. This procedure was done as an strategy to capture finer growth tendencies.

To rule out that a minor population of SmO is responsible for the multistep kinetic behavior observed for wildtype and A53T (Fig. 8c, d; T0), we performed additional kinetic traces for wildtype and A53T immediately after collecting pure monomeric αS from size exclusion runs and the same multistep profile was evidenced showing that the minor population of SmOs is not determinant for the observed behavior.

To generate seeds, we subjected the pellet of 1 mg mL⁻¹ mature fibrils to 15 s of ultrasonic exposure using an ultrasonic liquid processor (Misonix, #XL-2000). Previous to sonication, pellets were gently resuspended into 200 μL of the chosen condition to guarantee negligible amounts of monomers. We used a microprobe with a 3/32" tip diameter, and the output wattage was set to 15. Seeds were checked by TEM (Supplementary Fig. 8a) before incubation with 140 μM fresh monomers for ThT kinetic measurements. For the control experiments (Supplementary Fig. 8b–d), the solution of seeds was left at room temperature for ~90 h before checking for soluble and fibrillar fractions.

**Soluble and fibrillar measurements**. We measured the soluble and fibrillar fractions after subjecting 50 μL samples to ultracentrifugation (Beckman Coulter, #Optima MAX-TL) for 15 min 100,000 rpm at 20 °C. The supernatant and pellet fractions were immediately separated after centrifugation because some studied conditions produced a soft nonadherent pellet at the bottom of the tubes. We pipette from the top-center part of the supernatant to avoid any contamination with the fibrillar material. Dried pellets containing fibrils were vigorously homogenized with 50 μL of 5 M guanidine (USB, #50-01-1) to estimate the αS fibril concentration from the monomeric αS released from the fibrils after guanidine treatment. The concentrations of the soluble and fibrillar fractions were then measured by absorbance at 280 nm and molar extinction of 5960 M⁻¹ cm⁻¹ using a Nanodrop Onec system (Thermo Scientific Inc.).

**Transmission electron microscopy (TEM)**. We observed that a 5-fold dilution with water (20 μL of end kinetic products + 80 μL of MilliQ water) followed by gentle swirling using a vortexer led to better spreading of the αS fibrils on TEM grids. For conditions in which appreciable amounts of the soluble fraction were observed (e.g., A30P and E46K, −NaCl), the samples were not subjected to dilution. We prepared the soluble fraction grids (Supplementary Figs. 2 and 9a, b) after taking the supernatant (4 μL) of 50 μL samples previously subjected to 15 min at 100,000 rpm and 20 °C. For pellet grids (Fig. 8b and Supplementary Fig. 9c), we resuspended sedimented fibrils in 50 μL of water.

TEM images were obtained immediately after kinetic runs to confirm the formation of mature amyloid fibrils in the studied conditions (Fig. 1e). For length distribution measurements, images were taken at low magnification to guarantee

that both ends of longer filaments were seen within the micrographs. (Fig. 4a, b insets) All samples were applied to a previously discharged carbon film on 300 mesh copper grids (EMS, #CF300-Cu-Th) for 1-2 min, gently dried with filter paper and stained for 10 s with 2% uranyl acetate (EMS, #22400). Negatively stained samples were imaged on a Philips Tecnai microscope operated at 80 kV.

**Proteolytic digestion**. Mature αS fibrils (100 μM, ~1.4 mg mL$^{-1}$) under selected conditions (Fig. 2 and Supplementary Fig. 6) were treated at 37 °C with 4.4 μg mL$^{-1}$ PK (Thermo Scientific, #EO0491). Aliquots were taken over time (0, 1, 5, 10, 30, and 60 min) and immediately heated to 90 °C for 5 min in the presence of denaturing buffer to interrupt the cleavage reaction. The time course of cleavage reactions was monitored in a PhastGel system using 20% homogeneous SDS-PAGE (GE Lifesciences, #17-0624-01).

**Circular dichroism (CD)**. CD was carried out on a Jasco spectropolarimeter (J-1500) equipped with a Peltier temperature controller. Far-UV spectra were acquired from 260 to 185 nm at 25 °C using 0.2-nm steps, 1 mm bandwidth, and accumulation of three spectra. Fibrillar materials (63 μM) grown under selected conditions were centrifuged at $60,000 \times g$ for 10 min to separate remaining monomers. The pellet was gently resuspended in a less optically active buffer (15 mM KH$_2$PO$_4$, pH 7.0) to allow a lower wavelength cutoff (~185 nm)[61]. The buffer was extensively degassed prior to resuspension to eliminate oxygen that absorbs light at wavelengths lower than 200 nm. A demountable cuvette of 0.01 cm was used for all measurements. Mean residue ellipticities [Θ] in degrees cm$^2$ dmol$^{-1}$ were calculated using the equation $[\Theta] = l.c.10.n$, where Θ is the measured ellipticity in millidegrees, $n$ is the number of peptide bonds, $l$ is the path length in cm, and $c$ is the molar concentration.

**Electrospray ionization mass spectrometry (ESI-MS)**. Wildtype and A53T fibrils (~1 mg mL$^{-1}$) in the chosen conditions (−NaCl or 100 mM salt) were treated with PK for 10 min, followed by 5 min of enzyme inactivation at 95 °C. The samples were centrifuged for 10 min at 80,000 rpm and 10 °C and then frozen at −80 °C prior to MS evaluation. The LC-MS system consisted of a Thermo Electron Velos Pro Orbitrap ETD mass spectrometer with an Easy Spray ion source connected to a 15 cm ThermoFisher 3 μm C18 Easy Spray column (through precolumn).

One microliter (1:50 dilution with 3% acetic acid/water) of the sample was injected, and the peptides were eluted from the column by an acetonitrile/0.1 M acetic acid gradient at a flow rate of 0.3 μL/min over 1.5 h. The nanospray ion source was set at 2.3 kV. The digest was analyzed using the rapid switching capability of the instrument for acquiring a full scan mass spectrum (Orbitrap, 60 K resolution, AGC 9E5) to determine the peptide molecular weights, followed by 20 product ion spectra (IT, AGC 8E3) to determine the amino acid sequence in sequential scans. The data-dependent settings were repeat count 1, repeat duration 30, exclusion list 400, and exclusion duration 60.

The raw data file was processed using ThermoFisher Proteome Discoverer 1.4.1 using the SEQUEST algorithm against a database of two sequences for αS (wildtype and A53T, NCBI reference sequence: NP_001139527.1). The search parameters were parent mass 10 ppm, fragment masses 1.0 Da, no enzyme, oxidation Met, and acetylation N-terminus. The resulting processed data were displayed using Proteome Software Scaffold 4.8.1 with the following filters: xcorr (+1 > 1.8, +2 > 2.2, +3 > 2.5, +4 > 3.5), Peptide Prophet > 60%, Protein Prophet > 90%, and total spectra counts).

**Analytical size exclusion chromatography (A-Sec)**. Soluble fractions obtained during the ThT kinetic measurements (Fig. 8c, d) were directly injected into a Superose 6 Increase 10/300 GL (GE Lifesciences, #29-0915-96). All runs were performed in 10 mM Tris-Cl (pH 7.4) at a flow rate of 0.7 mL min$^{-1}$, and the absorbance was monitored at 280 nm using an ÄKTA Prime System (GE Lifesciences). The column was previously calibrated using thyroglobulin, 670 kD; γ-globulin, 158 kD; ovalbumin, 44 kD; myoglobin, 17 kD; and vitamin B12, 1.35 kD (Bio-Rad, #151-1901).

**Cryo-electron microscopy and image processing**. Mature fibrils obtained after ThT kinetic measurements were subjected to 5-fold dilution in the same specified buffer and gently homogenized in a vortexer. Samples (4 μL) were applied to lacey carbon grids (Ted Pella, #01895-F) previously discharged for 30 s with a Solarus 950 Advanced Plasma System (Gatan Inc.), blotted for 5 s using Whatman 1 filter paper (Whatman, #1001 055) using a force setting of 5, and vitrified using Vitrobot Mark IV (FEI, Inc.). The samples were loaded in a Titan Krios 300 kV and imaged with a Falcon III operating in linear mode using a 1.07 Å per pixel sampling. Images were acquired at 75,000× using defocus ranging from −1 to −3 μm at 0.5 μm steps, an exposure time of 2–3 s with 34 or 59 equal fractionation doses, respectively. The total dose ranged from 40–50 electrons per Å$^2$. Targets were selected using the EPU software. Images were motion corrected using Motion-Cor2[62], and defocus was estimated using the CTFFIND3 software[63]. Images containing poor contrast transfer function estimation and defocus values higher than −3 μm were not considered for further analysis. We boxed amyloid filaments using the e2helixboxer software of the EMAN suite[64]. Filament extraction and power spectrum calculations were performed using SPIDER scripts[65].

**Statistics and reproducibility**. Kinetic traces are expressed as the mean ± s.e.m. of several wells of at least three independent plate experiments. Sample size was set to a minimum of $n = 5$. At least two protein batches were used for kinetic experiments. Replicate is defined as plate wells.

Soluble and fibrillar fractions were measured from end products of kinetic traces. Data are expressed as the mean ± s.e.m. from several wells of at least two independent plate experiments. Sample size was set to a minimum of $n = 9$.

For cryo-EM and negative staining TEM, fibrils from two batches were imaged. Size exclusion chromatography and circular dichroism was performed twice.

**Reporting summary**. Further information on research design is available in the Nature Research Reporting Summary linked to this article.

## Data availability
All data and biological materials generated in this study are available from the corresponding author upon request. The reporting summary for this article is available as a Supplementary file. The source data underlying Figs. 1a–d, f–m, 2b, 5e, f, 6a–d, 7a–h, 8c, d, and Supplementary Figs. 3a–d, 4 are provided in Supplementary Data 1.

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

## Acknowledgements
This research was supported by the Pew Charitable Trusts Foundation to G.A.P.d.O as well as by the Carlos Chagas Filho Foundation for Research Support in the State of Rio de Janeiro (FAPERJ), grants 210.008/2018, 202840/2018, the National Council for Scientific and Technological Development (CNPq), and the National Institute of Science and Technology for Structural Biology and Bioimaging (INCT), grants 465395/2014-7 and 402321/2016-2 to J.L.S. We thank Nicholas E. Sherman and J.J. Park for conducting the mass spectrometry runs at the Biomolecular Analysis Facility, W.M. Keck Biomedical Mass Spectrometry Lab., University of Virginia. We also thank Edward H. Egelman for his helpful suggestions on the cryo-EM data and power spectrum interpretation.

## Author contributions
G.A.P.d.O. conceived the idea, designed the research, performed the experiments, analyzed the data, prepared the figures and wrote the paper with edits by J.L.S.

## Competing interests
The authors declare no competing interests.
