## [Peer Review File · Communications Biology]

Reviewers' comments:

Reviewer #1 (Remarks to the Author):

This manuscript reports on a study aimed at characterising various aggregate species on the pathway from monomeric to fibrillar protein for alpha-synuclein wild type and some of its disease-related variants.

While the study contains a range of interesting observations, no clear message and overall picture emerges from the work. It reads like the description of a series of disconnected observations. This is at least partly due to some very unclear and confusing language that the authors employ. At the end, I give some examples for expressions/sentences that are confusing or wrong.

Otherwise, there are a range of technical points I would like to see addressed, as well as points linked to the interpretation of the data.

Major points:

1) I would like to see a SDS-PAGE gel to confirm the purity of the asyn after production. It is very unusual to not subject the asyn to purification by AEC or SEC before use. Furthermore, the analytical SEC results show a high molecular weight shoulder of the monomer peak. This could correspond to covalent dimers, which can form either by dityrosine formation (Woerdehoff et al., JMB 2017), or by disulfide bridge formation between non codon optimised synuclein expressed in E.coli. It has been shown that a few percent of alpha-synuclein expressed from non-codon optimised plasmids contains a cysteine at the C-terminus, which can lead to covalent dimer formation. A SDS-PAGE gel under non-reducing conditions needs to be performed to rule this out.

2) The experiments with the different surfaces is not conclusive, as nucleation of alpha-synuclein fibrils also always happens at the air water interface (Campioni et al., JACS 2014). A53T could simply have a faster primary nucleation rate at the air water interface, rather than also nucleate at the low affinity polymer surface. Unless surface affinities are independently characterised, this possibility needs to be discussed.

3) "These results show that seeding reactions are dominated by secondary nucleation events"

This cannot be stated like that, as these experiments were performed under shaking conditions, where fragmentation plays an important role. Without a detailed kinetic analysis, it is impossible to conclude whether or not secondary nucleation contributes under shaking conditions.

4) The authors claim, based on the data in figure 6 c), that A53T features secondary nucleation at neutral pH. This is a strong claim and one would need to see more evidence. Positive curvature of aggregation kinetics under quiescent conditions is an indication, but no proof of the presence of secondary nucleation, as it can also stem from other effects, such as sedimentation, which can skew kinetic traces. Unless the authors are prepared to perform a full analysis along the lines of Gaspar et al. Quart. Rev. Biophys. 2017, I would recommend interpreting these data more carefully. It is very implausible that the strong pH-dependence of secondary nucleation found for the wild type is completely different, based on an amino acid substitution that does not change the charge state of the protein.

5) Overall, I find the discussion of the results very confusing and I have not managed to extract any clear message from the manuscript. I don't think it is my job to fix this here, but I would simply like to highlight one aspect, which is the discussion on the role of salt. Not only is the physico-chemical discussion confusing, because effects are discussed that usually require much higher salt concentrations (water structure etc.) but also the link to in vivo behaviour is very speculative and not realistic. If the authors had wanted to probe the role of physiologically relevant variations in salt concentrations and type, they should have studied the effect of calcium (which has been done extensively and it has been shown that it has dramatic effects on the aggregation of alpha-synuclein) or they should have investigated whether potassium has a different effect from sodium. The main difference between the salt concentrations inside and outside a cell are the differences in sodium vs. potassium concentrations, not so much the absolute ionic strength.

More minor points:

1) The equation of "stages" in kinetic traces with "species" makes no sense. Amyloid formation is not a sequential process, whereby one species follows the next, but rather different species can coexist at all times in different proportions.

2) "salt does not act as a surrogate of shear forces"

Salt and shaking will of course have very different effects, one would never expect to be able to substitute one by the other. The reason why shaking induces aggregation is not only the effect of shear, but also the effect on the species that form at the air water interface

(Campioni et al. JACS 2014).

3) "a solution composed exclusively of seeds contained similar amounts of species in the soluble and fibrillar fractions"

It is unclear whether this solution had been pre-purified, i.e. monomer had been removed and then left to re-equilibrate, or whether in all cases the seed solutions that the authors use simply contain 50% monomer already to start with. This needs to be clarified.

4) CD spectra in Figure 2: did the authors remove the monomer content before measurement by centrifugation and measurement of only the pellet?

5) Ionic strength variation (1,10,100 mM). It should be clearly mentioned somewhere in the text that the concentration of the buffer is already 10 mM, and that hence the addition of 1 mM NaCl hardly changes the ionic strength. What are the reaction conditions (shaking, type of recipient...)

Examples for unclear/incorrect formulations/language:

Abstract:

"continuously persists elongating fibrils in the presence of preformed seeds, and overcomes wild-type surface attributes for growth"

Even after reading the paper, I am not sure what that sentence means.

Introduction:

"More recently, α S polymorphs containing two staggered protofilaments has been achieved by cryo-EM."

This is not a complete sentence, something is missing here.

"been employed to underlie the commonalities and structural differences of inherited α S constructs."

underlie is not the right word here

"long-range flexibility of these filaments"

What is meant by long-range flexibility in the context of filaments?

"place A53T with new pathological advantages"

What is a pathological advantage? Do the authors mean a property that leads to more pronounced pathology? Then they should state it that way.

Results:

"Controversially, A30P and E46K fibrils formed in the presence of 1 mM of salt"

The authors probably mean conversely

"inherited α S fibrils"

What do the authors mean by that? do they mean "fibrils formed by the different sequences with heritable point mutations"? If so, they should express it correctly.

"polystyrene plates reliably elongate fibrils while a PEGylated surface does not"

This be written as "nucleate", rather than "elongate". Alpha-synuclein fibrils can grow in solution, but they can only nucleate on a surface.

Reviewer #2 (Remarks to the Author):

The current study has attempted to tease out fibril growth kinetics of alpha-synuclein and its mutant-variants in the presence of salt. It is an interesting study and several experiments were used towards this end. Salt has been well known to affect fibril formation but the authors show that certain ion concentrations promote multistep fibril formation. However, I have some concerns due to what I believe are incorrect assumptions. My comments and questions are as follows:

1) What were the objective criteria to determine two-step or three-step? For example, I find three-step for 1 mM E46K (one around 40 h), and similarly three-step for A53T 1 mM (one at 25 h).

- 2) Line 141. What data do the authors have to suggest that these fibrils have two protofilaments wrapping around? They could possibly collect STEM data.
- 3) Line 157-158: This statement is only true under the assumption that all fibrils pellet in a similar manner. Many fibrils pellet at different rates. This will affect the interpretation of data. As can be seen by TEM that some fibrils remain in solution even after ultracentrifugation. The selection of certain conditions (Lin1 172-176), therefore, would be based on a wrong assumption that is critical for interpreting the data.
- 4) Line 162: It is a mere assumption that E46K oligomers have more amyloid-like features. There is no data to support this claim. A similar assumption is made again regarding structure without any data.
- 5) Fig 2A: A30P pattern looks very similar to wt and A53T.
- Fig. 2B: The CD data seems to be obvious since A30P and E46K have fewer fibrils they show an increased signal for the disordered region as compared to wt/A53T. The wt and A53T spectrum should be shown separately. Or is it that wt/A53T overlap perfectly?

Reviewer #3 (Remarks to the Author):

This study presents an important step forward the understanding of mechanisms responsible for aggregation of α -synuclein into aggregates (fibrils, oligomers). α -synuclein, like other amyloid proteins, still remains a black box concerning the molecular basis of its aggregation. It's partially due to the fact that several groups have reported different α -synuclein polymorphs depending the conditions used for in vitro polymerization. On top of that, it has been difficult to obtain excellent quality kinetic data on such amyloid systems to extrapolate accurate mechanistic information. Numerous papers have been reported on α -synuclein aggregation, however there is a clear need to deepen our understanding.

The authors have focused on wt and the A53T mutant of α -synuclein and they've used various methods to characterize their aggregation products and investigate their aggregation kinetics. The force of the manuscript is the very high quality data obtained, to my knowledge it represents one of the most documented study on α -synuclein aggregation, especially on intermediate species. Original and technically very challenging is the last part of the study, attempting to reconstitute a time-course characterization using several techniques.

The proposed methodology results in excellent data and will undoubtedly lead to a better understanding of the early events associated with α -synuclein aggregation, which might in turn influence our way to design therapeutic strategies against the aggregation. The biological significance of their paper is important, because their results highlight differences of aggregation mechanisms between wt and the pathological variant A53T.

Their results are appealing, not only for the researchers working on α -synuclein but also for the more general amyloid community. Because it has been proposed that α -synuclein might also behave as a prion (at least the protein might have prion-like properties), the authors' work will also definitively be of interest for the prion community.

Therefore the work is interesting and timely and I recommend publication in Communications Biology.

Minor comments to be addressed:

- . Line 84: "the long-range flexibility" : not clear what the authors mean by flexibility. SSNMR structure determination is limited by static disorder of the samples, while cryo-EM structure determination by the heterogeneity at the macroscopic scale.
- . the authors should explain with more details at the beginning of the manuscript where these mutants come from and why they are important in the context of the disease propagation.
- . the authors describe in detail the effect of the ionic strength. What would be the nearest conditions to physiological condition?
- . Sample preparation for EM and cryo-EM analysis: a vortexing step and ultracentrifugation step are described, that might influence the observation of shorter/broken filaments, can the authors comment on this possibility?
- . Scale bars in Fig.1 and SI Fig.2 are too small, please increase the size.
- . Line 162 "amyloid-like features": no clear what the authors mean
- . Line 199-200: the 4.7Å spacing is a hallmark of cross-beta architecture, not "amyloid filaments" (amyloid fibrils can have a non cross-beta structure).

Reviewer#1:

1. While the study contains a range of interesting observations, no clear message and overall picture emerges from the work. It reads like the description of a series of disconnected observations. This is at least partly due to some very unclear and confusing language that the authors employ. At the end, I give some examples for expressions/sentences that are confusing or wrong.

Authors:

In this revised draft, we made substantial efforts to provide a more transparent language to the readers. We polished the text according to the reviewer comments. Please, track the changes in the enclosed highlighted file.

Reviewer #1:

2. I would like to see a SDS-PAGE gel to confirm the purity of the asyn after production. It is very unusual to not subject the asyn to purification by AEC or SEC before use. Furthermore, the analytical SEC results show a high molecular weight shoulder of the monomer peak. This could correspond to covalent dimers, which can form either by dityrosine formation (Woerdehoff et al., JMB 2017), or by disulfide bridge formation between non codon optimised synuclein expressed in E.coli. It has been shown that a few percent of alpha-synuclein expressed from non-codon optimised plasmids contains a cysteine at the C-terminus, which can lead to covalent dimer formation. A SDS-PAGE gel under non-reducing conditions needs to be performed to rule this out.

Authors:

In this improved version of the manuscript, we added a new Supplementary Fig.3 containing results of protein preps. After each batch preparation, we performed SDS-PAGE (in which runs contained 2-mercaptoethanol) and analytical SEC (Superdex 200) of an aliquot to check sample purity (new Suppl. Fig.3) prior to kinetic experiments. Based upon naked eye inspection of gels and because peak integration analysis of SEC chromatograms revealed > 95% of sample purity, we decided to proceed with no further chromatography step. We were very careful to judge both the sample purity and the use of fresh preps for kinetic experiments. All kinetic traces were performed with very fresh material (batches were no longer used after 6 months) because trace reproducibility was hardly achieved after longer freezing periods.

With respect to the “high molecular weight shoulder of the monomer peak”, this profile was not observed in Superdex 200 at the time of the quality checks (new Suppl Fig. 3). SEC runs in which the shoulder behavior was noticed were performed in a Superose 6 enhanced column that is optimized to resolve high MW species.

Following the reviewer request, we also performed a new SDS-PAGE under non-reducing conditions (in the absence of 2-mercaptoethanol) (new Suppl Fig. 3). We noticed a weak band over all studied constructs that may represent a small fraction of the covalent dimers raised by the reviewer. It is an important aspect to report, but

because all variants presented a small fraction of these potentially occurring dimers, we believe it does not substantially impact the comparative analysis of the kinetic traces we performed throughout the paper and the dissimilarities captured between the studied proteins.

Now read within the results section: “*Because of the intrinsically disordered nature of α S and its propensity to rapidly form higher order species, quality checks after sample preparation are provided in Supplementary Fig. 3.*”

Now read within the discussion section: “*A recent study showed that α S is prone to covalent dimerization triggered by dityrosine crosslinks.⁵⁹ However, it is still puzzling whether the observed multimers would represent this kind of α S covalent dimers.*”

Now read within the methods section: “*All kinetic traces were performed with fresh samples, ideally within 1-4 weeks after purification as reproducibility was hardly achieved with samples stored for longer periods (2-6 months) probably due to the accumulation of abnormal oligomeric species.*”

Reviewer #1:

3. The experiments with the different surfaces is not conclusive, as nucleation of alpha-synuclein fibrils also always happens at the air water interface (Campioni et al., JACS 2014). A53T could simply have a faster primary nucleation rate at the air water interface, rather than also nucleate at the low affinity polymer surface. Unless surface affinities are independently characterized, this possibility needs to be discussed.

Authors:

We acknowledge the reviewer for his/her comment. This is certainly an important point that raised our curiosity for future investigation. Indeed, alpha-synuclein growth kinetics represent a complex process of heterogeneous nucleation not only at the air water interface as raised by the reviewer, but also at the polymer-water interface and lipid-water interface. Further discussion about this topic was included in this improved version. Now read: “*Amyloid fibril formation of α S is dependent on a heterogeneous nucleation process at interfaces including the polymer-water interface,^{37,38} the air-water interface,³⁹ or lipid-water interface.⁴⁰ We evidenced that unlike wt, the A53T mutant was able to elongate fibrils even when incubated in a low adhesion surface for growth. This is an indicative that A53T has, to some extent, the ability to overcome the polymer-water interface requirement. It is true that A53T may compensate this by accelerating the primary nucleation at the air-water interface, but this is awaiting further exploration.*”.

Reviewer #1:

4. “These results show that seeding reactions are dominated by secondary nucleation events”. This cannot be stated like that, as these experiments were performed under shaking conditions, where fragmentation plays an important role. Without a detailed kinetic analysis, it is impossible to conclude whether or not secondary nucleation contributes under shaking conditions.

Authors: We tuned down this statement. Now read: *“Although fragmentation is present due to shaking, the seeding reactions suggest that secondary nucleation events may have a role on α S growth kinetics (i.e., dependence on the concentration of free monomers and the aggregate mass).”*

Alpha-synuclein presents a complex kinetics with heterogeneous nucleation process. As far as we know there is no model to analytically evaluate all steps of alpha-synuclein kinetic traces.

Reviewer #1:

5. The authors claim, based on the data in figure 6 c), that A53T features secondary nucleation at neutral pH. This is a strong claim and one would need to see more evidence. Positive curvature of aggregation kinetics under quiescent conditions is an indication, but no proof of the presence of secondary nucleation, as it can also stem from other effects, such as sedimentation, which can skew kinetic traces. Unless the authors are prepared to perform a full analysis along the lines of Gaspar et al. *Quart. Rev. Biophys.* 2017, I would recommend interpreting these data more carefully. It is very implausible that the strong pH-dependence of secondary nucleation found for the wild type is completely different, based on an amino acid substitution that does not change the charge state of the protein.

Authors: We tuned down our previous interpretation in this improved draft. Now read: *“This result is an indication that secondary nucleation processes might happen for this mutant at neutral pH, but because the A53T does not substantially modify the charge state of α S, we became more conservative to exclusively attribute secondary nucleation processes to the observed kinetic traces at neutral pH. Sedimentation would also explain to some extent, the observed effects. Notwithstanding, it is interesting because α S was previously shown to aggregate through secondary nucleation only at mildly acidic conditions, revealing a potential advantage of this mutant at physiological pH.”*

Reviewer #1:

6. Overall, I find the discussion of the results very confusing and I have not managed to extract any clear message from the manuscript. I don't think it is my job to fix this here, but I would simply like to highlight one aspect, which is the discussion on the role of salt. Not only is the physico-chemical discussion confusing, because effects are discussed that usually require much higher salt concentrations (water structure etc.) but also the link to in vivo behaviour is very speculative and not realistic. If the authors had wanted to probe the role of physiologically relevant variations in salt concentrations and type, they should have studied the effect of calcium (which has been done extensively and it has been shown that it has dramatic effects on the aggregation of alpha-synuclein) or they should have investigated whether potassium has a different effect from sodium. The main difference between the salt concentrations inside and outside a cell are the

differences in sodium vs. potassium concentrations, not so much the absolute ionic strength.

Authors:

We improved the text and tune down speculative claims, but it is the authors understanding that a discussion section should have some general interpretations and few speculations as a source of creativity and inspiration for future works. As stated in the former draft, we were aware of the gaps linking the in vitro findings to in vivo behavior. We have followed the reviewer suggestion and have polished this part of the discussion.

In addition, we do understand the caveat of further investigation of other ions (as mentioned, calcium and potassium). The criticism motivated someone in our group to explore the influence of these ions during a-syn growth kinetics, but it is authors understanding that further experiments on this is far beyond the scope of this manuscript and may be addressed in the future.

Reviewer #1:

1) The equation of “stages” in kinetic traces with “species” makes no sense. Amyloid formation is not a sequential process, whereby one species follows the next, but rather different species can coexist at all times in different proportions.

Authors: We clarified this point within the discussion section. Now read: “*We should stress that amyloid formation is not a process in which species are formed sequentially. Rather, population-weighted averages of multiple species coexist at each ThT stage in different proportions.*”

Reviewer #1:

2) “salt does not act as a surrogate of shear forces”

Salt and shaking will of course have very different effects, one would never expect to be able to substitute one by the other. The reason why shaking induces aggregation is not only the effect of shear, but also the effect on the species that form at the air water interface (Campioni et al. JACS 2014).

Authors: We recognize this was a naive and weak statement. In this revised draft, we excluded this statement. The effect of the air-water interface was addressed with further discussion.

Reviewer #1:

3) “a solution composed exclusively of seeds contained similar amounts of species in the soluble and fibrillar fractions”

It is unclear whether this solution had been pre-purified, i.e. monomer had been removed and then left to re-equilibrate, or whether in all cases the seed solutions that the authors use simply contain 50% monomer already to start with. This needs to be clarified.

Authors: The seed solution was produced from a pellet of mature fibrils after ultrasonic exposure, so the amount of monomers is negligible. Now read: *“To generate seeds, we subjected the pellet of 1 mg/mL mature fibrils in a chosen condition to 15 s of ultrasonic exposure using an ultrasonic liquid processor (Misonix, #XL-2000). Previous to sonication, pellets were gently resuspended into 200 μ L of the chosen condition to guarantee negligible amounts of monomers.”*

Reviewer #1:

4) CD spectra in Figure 2: did the authors remove the monomer content before measurement by centrifugation and measurement of only the pellet?

Authors: In this revised draft we provide a full description of how the CD measurements were performed. Now read: *“Fibrillar materials (63 μ M) grown under selected conditions were centrifuged at 60,000 xg for 10 min to eliminate remaining monomers.”*

Reviewer #1:

5) Ionic strength variation (1,10,100 mM). It should be clearly mentioned somewhere in the text that the concentration of the buffer is already 10 mM, and that hence the addition of 1 mM NaCl hardly changes the ionic strength. what are the reaction conditions (shaking, type of recipient...)

Authors: We included this information within the results section. Now read: *“Due to a buffer concentration of 10 mM (see Methods), the 1 mM NaCl condition has a small effect on ionic strength.”*

Reaction conditions were described in details within the Methods section.

“We used clear-bottom 96-well plates (Thermo Scientific, #265301) with a final reaction volume of 100 μ L in each well.”

“The temperature was 37°C throughout the entire kinetic measurement, and the orbital agitation was set to 10 s between reads unless stated otherwise.”

Reviewer #1:

Examples for unclear/incorrect formulations/language:

Abstract:

“continuously persists elongating fibrils in the presence of preformed seeds, and overcomes wild-type surface attributes for growth”

Even after reading the paper, I am not sure what that sentence means.

Authors:

We clarified the statements and provided further explanation. Now read: *“By showing that (i) A53T rapidly nucleates competent species for growth, (ii) continuously elongates fibrils in the presence of increasing amounts of seeds, and (iii) overcomes*

wild-type surface requirements for growth, our findings place A53T with new features that may explain the early onset of familial PD cases bearing this mutation.”

The first statement “*A53T rapidly nucleates competent species for growth*” refers to the results of Fig. 1a, d (black traces) and Fig. 8b, c, and d. In contrast to wt, these results evidence that A53T has faster kinetic traces (within the range of 10 to 30 h, Fig. 8b) and different oligomeric profiles (Fig. 8c, d). Although oligomer formation of either wt and A53T occurs within the same timeframe in the kinetic traces (between 4-10 h, Fig. 8b), A53T rapidly elongates fibrils (abrupt ThT binding within 10-30 h, Fig. 8b) while the wt did not. Because the SEC results evidenced different oligomeric compositions for wt and A53T, but oligomer formation occurs within the same timeframe for these proteins, the obvious conclusion is that A53T nucleates more competent species allowing a faster growth.

The second statement “*continuously elongates fibrils in the presence of increasing amounts of seeds*” refers to comparative results between Fig. 7f (check invariant bar) and Fig. 7h (check ascending bar). In contrast to A53T, wt is not able to increase the fibril mass in the presence of increasing amounts of seeds. This result has a clear conclusion that was better clarified in this revised draft, the one that A53T continuously elongates fibrils in the presence of increasing amounts of seeds.

The third statement “*overcomes wild-type surface requirements for growth*” refers to comparative analysis of Figs. 6d, e. When wt is grown in tubes containing low adhesion surfaces, the protein is not able to form fibrils. This observation is not true for the A53T because A53T fibrils are detected even though the protein is grown in low surface binding tubes. Therefore, wt has some surface requirements for growth that are bypassed in the case of A53T.

Reviewer #1:

Introduction:

“More recently, α S polymorphs containing two staggered protofilaments has been achieved by cryo-EM.”

This is not a complete sentence, something is missing here.

Authors: Now read “*More recently, α S polymorphs containing different quaternary arrangement and staggered protofilaments has been reported by cryo-EM.*”

Reviewer #1:

“been employed to underlie the commonalities and structural differences of inherited α S constructs.”

underlie is not the right word here

Authors: underlie > *uncover*

Reviewer #1:

“long-range flexibility of these filaments”

What is meant by long-range flexibility in the context of filaments?

Authors: During text polishing, we observed that the previous statement was not contributing for the main idea of the manuscript, so the sentence was deleted. For the curiosity of the reviewer, in cryo-EM structure determination we may think of filaments as 2D crystals, but different from 3D protein crystals in which long-range order is maintained, filaments do not preserve this order mainly due to thermal fluctuations along the filament axis. This observation challenges the generation of near-atomic cryo-EM maps of amyloid filaments.

Reviewer #1:

Results:

“Controversially, A30P and E46K fibrils formed in the presence of 1 mM of salt”

The authors probably mean conversely

Authors: Yes.

Reviewer #1:

“inherited α S fibrils”

What do the authors mean by that? do they mean “fibrils formed by the different sequences with heritable point mutations”? If so, they should express it correctly.

Authors: All sentences were corrected accordingly.

Reviewer #1:

“polystyrene plates reliably elongate fibrils while a PEGylated surface does not”

This be written as “nucleate”, rather than “elongate”. Alpha-synuclein fibrils can grow in solution, but they can only nucleate on a surface.

Authors: Corrected accordingly.

Reviewer #2 (Remarks to the Author):

The current study has attempted to tease out fibril growth kinetics of alpha-synuclein and its mutant-variants in the presence of salt. It is an interesting study and several experiments were used towards this end. Salt has been well known to affect fibril formation but the authors show that certain ion concentrations promote multistep fibril formation. However, I have some concerns due to what I believe are incorrect assumptions. My comments and questions are as follows:

1) What were the objective criteria to determine two-step or three-step? For example, I find three-step for 1 mM E46K (one around 40 h), and similarly three-step for A53T 1 mM (one at 25 h).

Authors: The criteria used to determine the transitions was naked eye inspection of the kinetic traces. We do understand the caveat of this non-objective strategy, but to our knowledge, there is no current kinetic model for α S to fit these complex curves and help on the identification of smooth transitions as those raised by the reviewer. Because of that, we limited our 'transition determination' to those clearly evident to the naked eye. Because alpha-synuclein amyloid formation is dependent on a heterogeneous nucleation process at interfaces, such as the polymer-water interface, the air-water interface or lipid-water interface, analytical methods to fit these complex curves are still awaiting further development.

2) Line 141. What data do the authors have to suggest that these fibrils have two protofilaments wrapping around? They could possibly collect STEM data.

Authors: We acquired TEM images using a Ceta16M camera that combines a large field of view with high speed readout. These features allowed us to image very clear micrographs and distinguish that the studied fibrils are formed by two intertwined protofilaments. Please, take a look Fig. 4e.

3) Line 157-158: This statement is only true under the assumption that all fibrils pellet in a similar manner. Many fibrils pellet at different rates. This will affect the interpretation of data. As can be seen by TEM that some fibrils remain in solution even after ultracentrifugation. The selection of certain conditions (Lin1 172-176), therefore, would be based on a wrong assumption that is critical for interpreting the data.

Authors: We have tuned down our statements. Nonetheless, we have some pieces of evidence to argue that selected conditions refer to those in which amyloid conversion is favored and the amount of fibrillar material within the corresponding soluble fractions is negligible: (i) we did several independent readings of soluble fractions (see Fig. 1f-i) to better approximate the observed effect; (ii) the presence of few outliers (see Fig. 1f-i) is a clear indication of healthy results and this observation by itself reports the reviewer complain - monomers, oligomers, and small protofibrils are probably present within the soluble fraction as shown in Suppl Fig. 2 and may give rise to the observed variance of the soluble fraction measurements; (iii) the fibrillar material observed by TEM within the soluble fractions (Suppl Fig. 2) was found after extensive screening of the grids and does not represent major species throughout the grid; (iv) their morphology reflect shorter and immature protofibrils bound to a great amount of oligomers (please, check Suppl Fig.2). We do not have a clear explanation of why these fibrillar species are present within the soluble fraction, but one hypothesis is the fact that because several soluble oligomers are close (or bound?) to these protofibrils, it might affect the pellet rate of the protofibrils, but again it represents a negligible population of protofibrils; (v)

we believe the longer and mature fibrils are denser and will form the major species of our pellets, so those conditions in which approximately 1mg/mL of soluble material were reached (and consequently ~ 1 mg/mL of fibrillar material) represent the equilibrium conditions in which amyloid formation was favored; (vi) selected conditions were checked by TEM and the amount of fibrillar material found into the grids is massive.

4) Line 162: It is a mere assumption that E46K oligomers have more amyloid-like features. There is no data to support this claim. A similar assumption is made again regarding structure without any data.

Authors: We tuned down our interpretation related to this point.

5) Fig 2A: A30P pattern looks very similar to wt and A53T.

Authors: We attached below the colored gels and arrows to highlight the dissimilarities between A30P, wt, and A53T.

Fig. 2B: The CD data seems to be obvious since A30P and E46K have fewer fibrils they show an increased signal for the disordered region as compared to wt/A53T. The wt and A53T spectrum should be shown separately. Or is it that wt/A53T overlap perfectly?

Authors: We realized the first draft did not include a full description of how we performed the CD data. In this revised draft, we performed additional CD measurements (new Fig.2b) and provide high-quality data together with a full description of the CD experiments. The main changes that allowed us to obtain higher-quality data were: (i) we used a cuvette containing a shorter path length and increased mass of fibrillar material to compensate the low signal-to-noise ratio at regions < 205 nm; (ii) we centrifuged the samples and gently resuspended the fibrillar material in a less optically active buffer; (iii) we standardized equal amounts of fibrillar mass to obtain more reliable comparisons, and (iv) the resuspension buffer was degassed to decrease the amount of light absorption below 200 nm due to the presence of oxygen gas.

Now read within the methods section: “CD was carried out on a Jasco spectropolarimeter (J-1500) equipped with a Peltier temperature controller. Far-UV

spectra were acquired from 260 to 185 nm at 25°C using 0.2-nm steps, 1 mm bandwidth, and accumulation of three spectra. Fibrillar materials (63 μM) grown under selected conditions were centrifuged at 60,000 xg for 10 min to eliminate remaining monomers. The pellet was gently resuspended in a less optically active buffer (15 mM KH₂PO₄, pH 7.0) to allow a lower wavelength cutoff (~ 185 nm).⁶¹ The buffer was extensively degassed prior to resuspension in order to eliminate oxygen that absorbs light at wavelengths lower than 200 nm. A demountable cuvette of 0.01 cm was used for all measurements. Mean residue ellipticities [Θ] in degrees.cm².dmol⁻¹ were calculated using the equation [Θ] = Θ /l.c.10.n, where Θ is the measured ellipticity in millidegrees, n is the number of peptide bonds, l is the path length in cm, and c is the molar concentration.”

Now read within the results section: “Circular dichroism (CD) data of the fibrillar materials revealed the typical amyloid β -sheet pattern containing negative bands at 218 nm and positive bands at 195 nm. The A53T fibrils revealed a dislocation of the positive band attributed to π - π^* amide group electronic transitions from 195 to 200-205 nm (Fig. 2b).”

Now read within the discussion section: “In contrast to the wt fibrils grown in the absence of salt and the A30P and E46K fibrils grown in the presence of 1 mM of salt, the far-UV spectrum of A53T fibrils grown in the absence of salt revealed a dislocated π - π^* electronic transition (positive band at ~ 200-205 nm) consistent with the presence of parallel β -sheets.⁴³ Due to the H-bond organization of parallel β -sheets, this secondary architecture is less stable than the anti-parallel sheet organization. This observation may allow to A53T monomers a faster screening for competent converting species at the early stages amyloid nucleation.”

Reviewer #3 (Remarks to the Author):

This study presents an important step forward the understanding of mechanisms responsible for aggregation of α -synuclein into aggregates (fibrils, oligomers). α -synuclein, like other amyloid proteins, still remains a black box concerning the molecular basis of its aggregation. It's partially due to the fact that several groups have reported different α -synuclein polymorphs depending the conditions used for in vitro polymerization. On top of that, it has been difficult to obtain excellent quality kinetic data on such amyloid systems to extrapolate accurate mechanistic information. Numerous papers have been reported on α -synuclein aggregation, however there is a clear need to deepen our understanding.

The authors have focused on wt and the A53T mutant of α -synuclein and they've used various methods to characterize their aggregation products and investigate their aggregation kinetics. The force of the manuscript is the very high quality data obtained, to my knowledge it represents one of the most documented study on α -synuclein aggregation, especially on intermediate species. Original and technically very challenging is the last part of the study, attempting to reconstitute a time-course characterization using several techniques.

The proposed methodology results in excellent data and will undoubtedly lead to a better understanding of the early events associated with a-synuclein aggregation, which might in turn influence our way to design therapeutic strategies against the aggregation. The biological significance of their paper is important, because their results highlight differences of aggregation mechanisms between wt and the pathological variant A53T.

Their results are appealing, not only for the researchers working on a-synuclein but also for the more general amyloid community. Because it has been proposed that a-synuclein might also behave as a prion (at least the protein might have prion-like properties), the authors' work will also definitively be of interest for the prion community.

Therefore the work is interesting and timely and I recommend publication in Communications Biology.

Minor comments to be addressed:

. Line 84: "the long-range flexibility" : not clear what the authors mean by flexibility. SSNMR structure determination is limited by static disorder of the samples, while cryo-EM structure determination by the heterogeneity at the macroscopic scale.

Authors: We polished this part of the text as we realized it was not adding to much to the manuscript.

the authors should explain with more details at the beginning of the manuscript where these mutants come from and why they are important in the context of the disease propagation.

Authors: We included additional information about the origins of these mutations. Now read: *“For instance, the A53T mutation was first reported in an Italian kindred and in three unrelated Greek families with autosomal dominant inheritance for PD. Further, the A30P was identified in three individuals of German origin. All individuals presented an onset of illness ranging from the mid-30s to the mid-50s.”*

the authors describe in detail the effect of the ionic strength. What would be the nearest conditions to physiological condition?

Authors: A sentence and a reference within the first paragraph of the results section show what would be the physiological concentration of Na⁺ across the membrane of neurons based upon MRI measurements (Ref 35). Read: *“We chose log scale salt increments (e.g., 1, 10, and 100 mM) to be fairly consistent with the physiological gradient of Na⁺ observed across the membrane of neurons (~10-15 mM intracellular against ~140 mM extracellular)³⁵.”*

Sample preparation for EM and cryo-EM analysis: a vortexing step and ultracentrifugation step are described, that might influence the observation of shorter/broken filaments, can the authors comment on this possibility?

Authors: We performed gentle vortexing and double checks by TEM after ultracentrifugation with no changes on filaments length.

Scale bars in Fig.1 and SI Fig.2 are too small, please increase the size.

Authors: Done.

Line 162 "amyloid-like features": no clear what the authors mean

Authors: We corrected this accordingly.

Line 199-200: the 4.7Å spacing is a hallmark of cross-beta architecture, not "amyloid filaments" (amyloid fibrils can have a non cross-beta structure).

Authors: We corrected this accordingly.

Sincerely,

REVIEWERS' COMMENTS:

Reviewer #1 (Remarks to the Author):

I think by reformulating many critical passages in the manuscript, the authors have tuned down the statements in the manuscript to a level compatible with the data. I have no more major objections against publishing it in Communications Biology.

P.S.: The authors should cite the dityrosine study, rather than just mentioning it.

Reviewer #2 (Remarks to the Author):

The authors have addressed all the comments satisfactorily. The current version would be acceptable.

Reviewer #3 (Remarks to the Author):

The authors have addressed the questions from the reviewers.

Authors reply to reviewer #1:

The dityrosine study was cited in the previous revised draft, ref. 59.